# NutriBench: A Dataset for Evaluating Large Language Models on Nutrition Estimation from Meal Descriptions

**Andong Hua**[1*]     **Mehak Preet Dhaliwal**[1*]     **Laya Pullela**[1]     **Ryan Burke**     **Yao Qin**[1]
[1]University of California, Santa Barbara
{dongx1997,mdhaliwal,lpullela,yaoqin}@ucsb.edu, ryan.e.burke@gmail.com

## Abstract

Accurate nutrition estimation helps people make informed dietary choices and is essential in the prevention of serious health complications. We present Nu-triBench, the first publicly available natural language meal description nutrition benchmark. NutriBench consists of 11,857 meal descriptions generated from real-world global dietary intake data. The data is human-verified and annotated with macro-nutrient labels, including carbohydrates, proteins, fats, and calories. We conduct an extensive evaluation of NutriBench on the task of carbohydrate estimation, testing twelve leading Large Language Models (LLMs), including GPT-4o, Llama3.1, Qwen2, Gemma2, and OpenBioLLM models, using standard, Chain-of-Thought and Retrieval-Augmented Generation strategies. Additionally, we present a study involving professional nutritionists, finding that LLMs can provide comparable but significantly faster estimates. Finally, we perform a real-world risk assessment by simulating the effect of carbohydrate predictions on the blood glucose levels of individuals with diabetes. Our work highlights the opportunities and challenges of using LLMs for nutrition estimation, demonstrating their potential to aid professionals and laypersons and improve health outcomes. Our benchmark is publicly available at: https://mehak126.github.io/nutribench.html

## 1 Introduction

Effective nutrition monitoring and dietary management are essential components of healthcare, closely linked to the prevention and control of chronic diseases, including obesity, heart disease, diabetes, and cancer (Amiri et al., 2019; Min et al., 2019; Rollo et al., 2016). For instance, individuals with diabetes must accurately estimate the carbohydrate content of meals to determine appropriate insulin doses (Contreras et al., 2023; Buck et al., 2022). Inaccurate carbohydrate estimation can result in high blood sugar (hyperglycemia) or low blood sugar (hypoglycemia), both of which can lead to severe short- and long-term health complications (Cologne & in Health Care , IQWiG; Sabarathinam, 2023).

Despite technological advancements in dietary assessment methods, nutrition estimation from self-reported dietary intake has shown limited accuracy and high user burden (Rollo et al., 2016; Taylor et al., 2021; Tosi et al., 2021). A major limitation is that most modern nutrition datasets typically include tabular data (USDA, 2019; Nutrition, 2018; Li, 2019; MenuStat; Open Food Facts; FoodDB, The Metabolomics Innovation Centre) or meal images paired with nutrition information (Contreras et al., 2023; Ruede et al., 2021; Liang & Li; Tai et al., 2023; Thames et al., 2021; Nair et al., 2023; Ma et al., 2023) but often lack natural language descriptions, restricting their usage and flexibility. For example, tabular database searches generally require an exact match for successful retrieval and multiple searches for meals with multiple food items, making the process time-consuming and burdensome (Hezarjaribi et al., 2017). In addition, image processing-based nutrition estimation systems are restricted to real-time predictions, pose privacy concerns (Hezarjaribi et al., 2017) and may encounter issues with food components being obscured in the image.

---

*Equal contribution, alphabetically ordered.

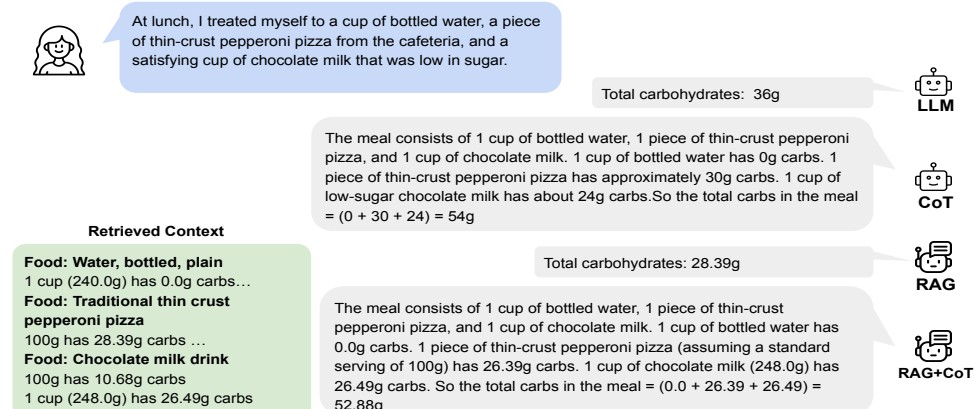

Figure 1: GPT-4o-mini answers a query from NUTRIBENCH using different prompting strategies.

In contrast, using natural everyday language to describe meals provides greater flexibility, enabling users to detail various meal components and serving sizes without the constraints of real-time input. This also allows users to record meals in advance for planning or retrospectively for tracking. We propose that Large Language Models (LLMs) are valuable tools for nutrition estimation from natural language meal descriptions due to their advanced language understanding and reasoning capabilities, vast internal knowledge and ability to refer to external sources to provide precise nutrition estimates. However, there are currently no existing datasets available to evaluate LLMs on this task.

To bridge this gap, we present NUTRIBENCH, a dataset comprising 11,857 natural language meal descriptions generated by GPT-4o-mini, derived from real-world global dietary intake data. NU-TRIBENCH is human-verified and annotated with macronutrient labels, including carbohydrates, proteins, fats, and calories. To our knowledge, NUTRIBENCH is the first publicly available benchmark for evaluating the performance of LLMs on nutrition estimation from meal descriptions. NU-TRIBENCH is constructed from dietary intake data from 11 countries, sourced from the What We Eat in America (United States Department of Agriculture) and the FAO/WHO GIFT (Leclercq et al., 2019) datasets. We generate meal descriptions that include single and multiple food items, specifying serving quantities both in precise metric measurements (such as grams) and in natural language terms (such as 'a tablespoon' or 'half a cup') to capture the variety of meal description styles used in the real world.

We evaluate 12 leading LLMs, including open-source models (Llama 3.1 (Dubey et al., 2024), Llama 3 (Meta, 2024), Gemma 2 (Team et al., 2024), and the Qwen 2 (Yang et al., 2024) family), closed-source models (GPT-4o and GPT-4o mini (Achiam et al., 2023)), and a medical domain-specific model (OpenBioLLM-70B (Ankit Pal, 2024)), on the task of carbohydrate estimation from natural language meal descriptions. The evaluation spans four prompting paradigms, including Chain-of-Thought (CoT) (Wei et al., 2022) and Retrieval-Augmented Generation (RAG) (Lewis et al., 2020). Figure 1 shows an example of GPT-4o-mini's output across different prompting strategies on a meal description from NUTRIBENCH.

Additionally, we conduct a study with three professional nutritionists to obtain carbohydrate estimates on a sample of meal descriptions from NUTRIBENCH. Figure 2 summarizes the performance of the evaluated LLMs and the nutritionists. GPT-4o with CoT prompting achieves the highest accuracy of 66.82% with 99.16% of the queries answered. Most LLMs provide more accurate and faster estimates than the nutritionists, highlighting their potential as valuable tools for obtaining precise and accessible nutrition estimates from natural language meal descriptions.

Finally, we performed a real-world risk assessment by simulating the effect of carbohydrate estimates on the blood glucose levels of Type 1 diabetes patients. An evaluation of blood glucose risk metrics across 44,800 simulations showed that nutrition estimates provided by LLMs can help Type 1 diabetes patients maintain their blood glucose within safe limits. Our experiments demonstrate the potential of LLMs to offer accurate dietary guidance, benefiting both professionals and laypersons and ultimately improving health outcomes.

Our contributions can be summarized as follows:

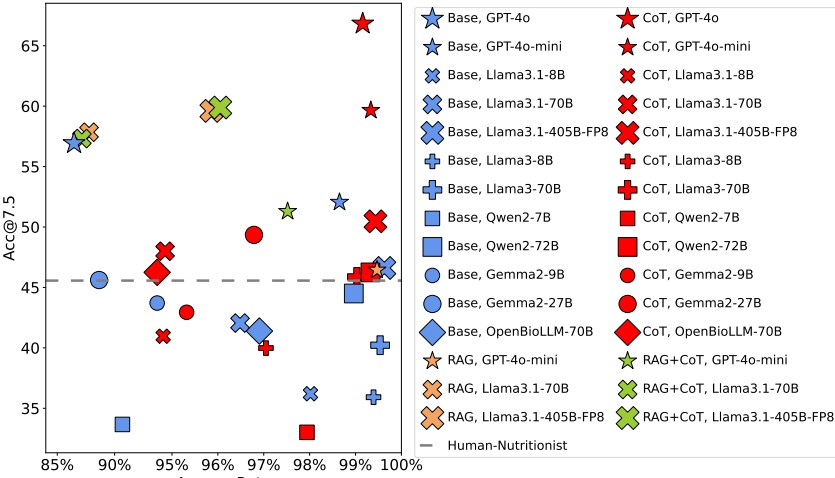

Figure 2: Accuracy and Answer Rate (defined in Section 4) for the evaluated LLMs across different prompting strategies on NUTRIBENCH.

- **NUTRIBENCH**: We present NUTRIBENCH - the first publicly available natural language meal description dataset labeled with macronutrient and calorie estimates. NUTRIBENCH consists of 11,857 human-verified meal descriptions derived from real-world dietary intake data from *eleven* countries spanning across the world.

- **Benchmarking LLMs**: We evaluate *twelve* LLMs varying in size and expertise covering both open and closed source models with different prompting strategies including CoT and RAG on the task of carbohydrate estimation on NUTRIBENCH. This provides a comprehensive insight into the current capabilities of LLMs in nutrition estimation.

- **Expert Nutritionist Evaluation:** We conducted a study with *three* professional nutritionists who provided carbohydrate estimates for a subset of meal descriptions from NUTRIBENCH and measured the accuracy and speed of their predictions against those of the LLMs.

- **Real-World Risk Assessment:** We conducted a study simulating the effect of meal carbohydrate estimation on blood glucose levels of individuals with type 1 diabetes, demonstrating the real-world impact and potential health benefits of LLM-driven nutrition guidance.

## 2 RELATED WORK

**Nutrition Estimation.** Most mainstream nutrition datasets such as FoodData Central (FDC) (USDA, 2019), MenuStat (MenuStat), FoodCom (Li, 2019), and Nutritionix (Nutrition, 2018) feature tabular food nutrition information curated from diverse sources. However, tabular datasets based retrieval methods require users to use precise terminology to ensure successful and relevant retrieval. In addition, these methods do not support retrieving multiple items that may be present in a meal in a single search, making the process time-consuming and burdensome (Hezarjaribi et al., 2017).

Another popular method for nutrition estimation involves predicting nutritional values from food images. Such approaches may involve identifying food items followed by retrieval from a tabular nutrition database (Yunus et al., 2018) or directly estimating the meal's nutritional breakdown from the image (Keller et al., 2024). Many existing datasets contain images paired with nutrition information to facilitate research in this direction. These datasets may include images obtained from the web (Ruede et al., 2021), real world, (Liang & Li; Tai et al., 2023; Thames et al., 2021), or synthetically generated images (Nair et al., 2023). UMDFood90k (Ma et al., 2023) provide a multimodel dataset with product images, text-based ingredient statements, and nutrient amounts. However, image-based nutrition approaches are time-sensitive (Hezarjaribi et al., 2017), requiring users to capture specific pictures of their meals at the time of consumption and may encounter issues with food components being obscured in the image.

Enabling users to input meal descriptions in natural everyday language can help mitigate these issues. Initial exploration in this realm includes (Korpusik et al., 2017; Taylor et al., 2021), who

use Convolutional Neural Networks to match food items from crowdsourced meal descriptions with an external tabular nutrition database. However, their data is not publicly released, preventing the evaluation of current state-of-the-art language processing approaches, such as LLMs, for this task.

Motivated by the lack of a standardized benchmark for nutrition estimation based on natural language meal descriptions, we introduce NUTRIBENCH, the first such publicly available benchmark, and evaluate state-of-the-art LLMs to gain insight into their current capabilities and limitations.

**Large Language Models (LLMs).** Large Language Models (LLMs) have made significant progress in recent years, with both closed-source models like GPT-4 (Achiam et al., 2023) and open-source models like Llama 3 and 3.1 (Dubey et al., 2024), Gemma 2 (Team et al., 2024), and OpenBi-oLLMAnkit Pal (2024) enabling advancements in natural language processing as well as knowledge-intensive, reasoning, and cross-domain tasks including healthcare applications (Gramopadhye et al., 2024). Despite their extensive internal knowledge, LLMs still suffer from issues such as hallucinations, incorrect or outdated information, and a lack of interpretability (Lin et al., 2021; Li et al., 2023; Zhao et al., 2024).

Chain-of-Thought (CoT) (Wei et al., 2022) prompting alleviates some of these issues by enabling models to reason about the answer step-by-step. Another promising solution is Retrieval-Augmented Generation (RAG) (Lewis et al., 2020), which provides the model with additional context relevant to the query by retrieving information from an external knowledge source. While previous works have identified shortcomings of both these approaches (Chen et al., 2024; Yao et al., 2024), they have not been assessed on the task of nutrition estimation, which may uncover unique challenges.

Our work comprehensively evaluates 12 state-of-the-art LLMs with standard, CoT, RAG, and RAG combined with CoT prompting for carbohydrate estimation from natural language meal descriptions. We present a detailed analysis of how different prompting strategies affect LLMs' performance in nutrition estimation in Section 5.

## 3 NUTRIBENCH CONSTRUCTION

NUTRIBENCH is derived from two source databases: What We Eat in America (United States Department of Agriculture) and FAO/WHO GIFT (Leclercq et al., 2019), which collectively provide real-world dietary intake data from 11 countries, including Argentina, Bulgaria, Ethiopia, India, Italy, Mexico, Nigeria, Peru, Philippines, Sri Lanka, and the United States. After processing the meal data with nutrition labels, we instruct GPT-4o-mini to generate natural language meal descriptions to construct NUTRIBENCH. Examples of the meal descriptions can be found in Appendix A.

### 3.1 WHAT WE EAT IN AMERICA

The What We Eat in America (WWEIA) survey is a national food survey jointly conducted by the U.S. Department of Health and Human Services (HHS) and the U.S. Department of Agriculture (USDA). Trained interviewers collect dietary intake data, including food names and portions through two 24-hour dietary recalls. As of September 2024, WWEIA has completed 10 surveys from 2001 to 2020, compiling a total of 2,070,592 food items. To obtain nutrition labels, we utilize the Food and Nutrient Database for Dietary Studies (Montville et al., 2013), a comprehensive database that provides nutrient values for foods and beverages within the WWEIA data.

We cleaned the WWEIA data by removing entries with missing or invalid information (e.g., "Quantity not specified" portion sizes) and integrated data across the years into one comprehensive dataset. Since both the recorded food names and nutrition labels from the FNDDS change across the years, we use the 2017-2018 dataset as our reference point[1]. Nutritional labels for each food item are calculated and normalized based on nutrition information derived from this reference data. Entries from the WWEIA that contain food names not found in the reference dataset have been excluded from our analysis.

### 3.2 FAO/WHO GIFT

The FAO/WHO GIFT (Leclercq et al., 2019) is an open-access online platform by the Food and Agriculture Organization of the United Nations that provides individual food consumption data compiled from global national and sub-national food consumption surveys.

---

[1]The FNDDS 2017-2018 dataset is the most recent available data that aligns with the WWEIA survey prior to the pandemic.

We compiled meals from 10 countries based on the availability of indicators that allowed us to separate food items into distinct meals (such as eating timestamps and meal markers), nutritional data, and geographical diversity. The countries are displayed in Figure 3. For each meal, we obtained the food items, their quantities in grams, and corresponding nutritional information. Since we aim to generate natural language meal descriptions with serving sizes expressed in everyday terms, we converted the food quantities from grams to common serving units. This was done by mapping the food items in the FAO/WHO GIFT dataset to The FoodData Central (FDC) (USDA, 2019) dataset and applying a rule-based algorithm (Appendix F) to translate the amounts in grams into natural serving sizes. FDC is the food composition information center of USDA (Fukagawa et al., 2022) providing detailed information on foods, including conversions from grams to natural servings. To avoid bias, we also considered meals where at least one item could not be mapped to an FDC item, keeping the original gram amounts.

**Combining Food Items to Meals**   Food items in WWEIA or FAO/WHO GIFT are recorded independently. Therefore, we merge food items into a single meal if they are consumed by the same person on the same day during the same eating occasion. For instance, a meal may include "a slice of buttered toast, a cup of black coffee, and half a grapefruit" consumed together. For each meal, we maintain either natural or metric serving sizes—ensuring there is no mix-up between the two. We compile 5,532 meal descriptions with natural servings, alongside the same 5,532 descriptions converted to metric servings from the WWEIA database. Additionally, Table 1 summarizes the number of meals and unique food items sampled from each country from the FAO/WHO GIFT.

| Country | Meals | Food items |
|---|---|---|
| (1) Argentina | 70 | 67 |
| (2) Bulgaria | 26 | 39 |
| (3) Ethiopia | 22 | 20 |
| (4) India | 18 | 16 |
| (5) Italy | 141 | 124 |
| (6) Mexico | 181 | 254 |
| (7) Nigeria | 124 | 86 |
| (8) Peru | 93 | 92 |
| (9) Philippines | 84 | 104 |
| (10) Sri Lanka | 34 | 43 |

Table 1: Unique meals and food items sampled from each country from FAO/WHO GIFT

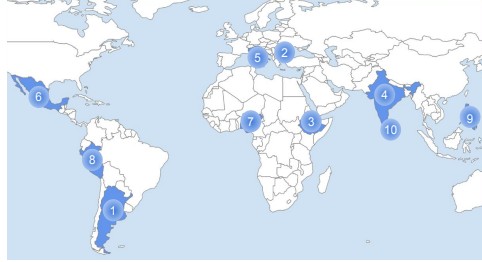

Figure 3: The countries from which meals were extracted from the FAO/WHO GIFT. The indices on the map correspond to the countries in Table 1.

### 3.3 GENERATING NATURAL LANGUAGE MEAL DESCRIPTIONS

**GPT-4o-mini Based Generation.** To obtain natural language meal descriptions from the collected meal data at scale, we instruct GPT-4o-mini to generate these descriptions, which are then used to create NUTRIBENCH . To encourage diversity, we prompt the LLM to produce five varied meal descriptions for each food item in a single generation, from which we randomly select one as the final query. Details of the prompts used are in Appendix B.1.

**Human Verification.** Human verification is conducted to ensure that the meal descriptions generated by the LLM contain no hallucinations or missing information. One of the authors acts as the verifier, confirming that the food names and serving sizes are accurately included in the final meal descriptions to match the nutrition labels. We identified two common mistakes made by GPT-4o-mini: missing food names and missing food servings. Additionally, we observed that GPT-4o-mini has a higher probability of omitting food names when a meal contains multiple food items and tends to make more mistakes generating descriptions with natural serving sizes compared to metric serving sizes. For example, a meal description prior to human verification, "During lunch, I had a tasty medium crust pepperoni pizza and washed it down with a carton of 100% orange juice." was updated to, "During lunch, I had a piece of tasty medium crust pepperoni pizza and washed it down with a carton of 100% orange juice.", after verification (note the change from a whole pizza to a single piece). More examples of this verification process are displayed in Appendix B.2.

## 4 EXPERIMENTS

**LLM Models**   We conduct a comprehensive evaluation of *twelve* state-of-the-art large language models (LLMs) using NUTRIBENCH on the task of carbohydrate estimation from meal descriptions. The evaluation spans a range of open-source models, including Llama 3.1-8B, Llama 3.1-70B, Llama 3.1-405B-FP8, Llama 3-8B, Llama 3-70B (Meta, 2024; Dubey et al., 2024), Gemma 2-9B, Gemma 2-27B (Team et al., 2024), and Qwen 2-7B, Qwen 2-72B (Yang et al., 2024). We also include closed-source models such as GPT-4o and GPT-4o mini (Achiam et al., 2023), as well as the medical domain-specific model OpenBioLLM-70B (Ankit Pal, 2024).

**Prompting Methods**   We adapted *four* existing prompting methods with carefully designed prompts tailored for carbohydrate estimation.

- **Base**: The first baseline prompt involves instructing LLMs to estimate the carbohydrate content based on the meal description provided in the query with basic instructions.

- **Chain-of-Thought (CoT)** (Wei et al., 2022): Due to the complexity of meal descriptions with multiple items in varying quantities, we hypothesize that the step-by-step reasoning induced by chain-of-thought prompting would reduce model errors by enabling the model to identify and reason about individual query components required to make the overall decision.

- **Retrieval-Augmented Generation (RAG)** (Lewis et al., 2020): To further enhance the reliability of LLM predictions, we use RAG to ground their generations with nutrition knowledge retrieved from a nutrition database. First, for a given meal query, we prompt the model to parse it into individual food components. Next, we retrieve nutrition information about each food item in the query through a nearest neighbor semantic similarity search and concatenate the results to form a comprehensive set of facts about the food components in the query. Finally, we provide this retrieved context along with the original prompts for LLMs.

- **RAG+CoT**: We combine the nutrition retrieval capability of RAG with step-by-step reasoning in CoT by concatenating the retrieved nutrition context with the CoT prompting for LLMs.

In all the cases above, we instruct the models to respond with '-1' if they don't know the answer to reduce the risk of potentially harmful predictions.

Figure 1 shows the predictions generated by GPT-4o-mini using the four prompting paradigms. We evaluate RAG and RAG+CoT with GPT-4o-mini, Llama3.1-70B, and Llama3.1-405B-FP8 only due to computation constraints. The prompts used for each method and hyperparameters can be found in Appendix C.

**Evaluation Metrics**   We calculate the mean absolute error (MAE) to measure the deviation of the model responses from the true meal carbohydrates. In addition, we report accuracy (Acc@7.5) by considering the model output as 'correct' if the predicted value is within ±7.5g of the ground truth. This is based on the insulin-to-carbohydrate ratio, which indicates the grams of carbohydrates one unit of insulin can cover. While this ratio varies among individuals, it is generally considered 1:15 as a rule of thumb [2]. Since we aim to improve insulin management, we maintain a conservative threshold of 7.5g on the absolute error to measure accuracy.

Finally, since we allow the models not to provide an estimate if uncertain, we also report the answer rate (AR), indicating the percentage of answered questions. Overall, the models should have a high answer rate and Acc@7.5, and a low MAE.

## 5 RESULTS

Figure 2 presents the accuracy and answer rate of the 12 evaluated large language models across four prompting paradigms. Across all models and methods, GPT-4o with Chain-of-Thought (CoT) prompting achieves the highest accuracy of 66.82%, with an answer rate of 99.16%. Among open-source models, Llama 3.1-405B-FP8 with RAG+CoT prompting performs the best, with an accuracy of 59.89% and an answer rate of 96.05%.

---

[2]https://www.tidepool.org/blog/optimizing-insulin-to-carb-ratios

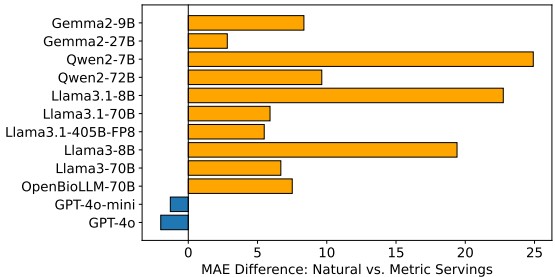
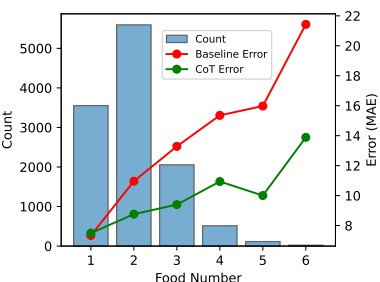

Figure 4: Difference in MAE between natural and metric serving descriptions. Natural servings are preferred by most models (orange), whereas the GPT family prefers metric (blue).

Figure 5: Frequency of queries and model error with increasing number of foods in descriptions.

### 5.1 IMPACT OF SERVING DESCRIPTIONS, PROMPTING STRATEGIES, AND RAG

**Natural Serving Descriptions are Preferred by Most Models.** We compare the performance of models on NUTRIBENCH with natural versus metric serving descriptions. Figure 4 shows the difference in MAE for each model between natural and metric serving descriptions. We observe that most models exhibit higher MAE when using natural serving measures (e.g., "a cup of rice"), compared to metric measures (e.g., "80g of rice"). Notably, both GPT-4o and GPT-4o mini outperformed in the metric serving category. We hypothesize that this may be attributed to the inclusion of nutrition information associated with metric servings in the training data for the GPT family.

**CoT Improves both Answer Rate and Accuracy, Especially on Complex Meal Descriptions.** Table 2 presents the answer rate and accuracy across the twelve models on NUTRIBENCH using Base and CoT prompting. We observe that CoT prompting significantly improves performance across both metrics. Furthermore, while larger models generally outperform their smaller counterparts, as shown in Figure 2, CoT prompting allows smaller models to exceed the performance of larger ones. For example, GPT-4o mini with CoT prompting achieves higher accuracy and answer rate than GPT-4o with Base prompting. This indicates that the step-by-step reasoning facilitated by CoT prompting improves the accuracy, demonstrating that even smaller models can achieve competitive or superior performance when paired with effective prompting strategies.

Looking more closely, we discover that CoT prompting helps mitigate increasing errors in complex meal descriptions. Figure 5 presents a histogram of the meal descriptions in NUTRIBENCH with increasing food items and the corresponding MAE for GPT-4o under Base prompting (red) and CoT prompting (green). There is a sharp increase in the MAE with an increasing number of food

Table 2: Accuracy and answer rate averaged over 12 models for Base and CoT.

|  | Base | CoT |
| --- | --- | --- |
| Acc @ 7.5 | 43.24% | 47.46% |
| Answer Rate | 95.59% | 97.14% |

items in the meal for Base prompting, CoT prompting significantly mitigates the error. This analysis demonstrates that enabling the model to parse meal descriptions into individual food items and perform explicit calculations through step-by-step reasoning improves the accuracy of LLMs, particularly for more complex queries.

**RAG Does Not Always Improve Performance.** In our experiments with Retrieval-Augmented Generation (RAG), we built a retrieval database, RETRI-DB, using data from FDC (USDA, 2019). First, we parsed meal descriptions into individual food items and retrieved relevant nutritional information from RETRI-DB. This data was then used as context to prompt the LLMs for carbohydrate estimation. Figure 1 shows an example of nutritional information retrieved for two food items in the input query. Further details on RETRI-DB construction are in Appendix D.

We evaluate the impact of RAG on GPT-4o-mini, Llama3.1-405B-FP8, and Llama3.1-70B. Figure 6 shows the accuracy and answer rates across four prompting methods for queries involving natural and metric serving descriptions. For metric-serving queries, RAG and RAG+CoT significantly outperform the Base and CoT for Llama3.1, indicating that the additional information retrieved for RAG provides valuable context that improves the accuracy of their predictions. However, for GPT-4o-mini, RAG shows only minor improvement over Base, while RAG+CoT performs worse than CoT. Similar to our hypothesis regarding the impact of natural versus metric serving sizes, we hy-

pothesize that this may be due to metric serving nutrition information being included in the training of GPT-4o models. Therefore, RAG may not provide additional information to improve outputs but could misguide the original prediction by potentially noisy retrieval results.

On the other hand, for natural serving queries, while RAG shows some improvement over Base for the Llama 3.1 models, CoT-only always outperforms RAG+CoT. An inspection of RETRI-DB reveals that for most food items, the nutritional data available in the retrieved context is for standardized metric 100g servings. We hypothesize

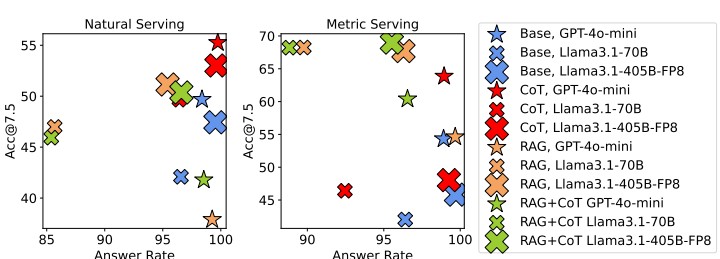

Figure 6: Comparison of model performance with RAG on natural and metric serving queries

that the LLMs are unable to successfully utilize this information when the food amounts in the query are described using natural terms. This suggests that alignment between the RAG database and the queries is crucial for improving results. A promising future direction is to augment RETRI-DB with more natural servings for RAG to support knowledge-grounded predictions with LLMs.

## 5.2 VARIATION IN LLM PERFORMANCE ACROSS DIETS AND CULTURES

We analyze whether there are differences in the performance of LLMs based on dietary patterns and cultural contexts.

**Correlation between meal carbohydrates and prediction error.** Figure 7 (*Left*) examines the relationship between meal carbohydrate content and prediction error. We plot the frequency and MAE by GPT-4o with CoT prompting for varying carbo-

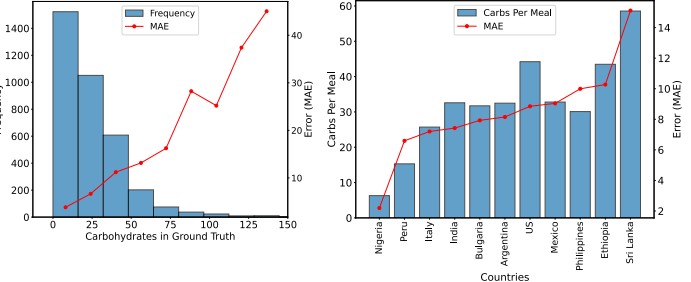

Figure 7: **Left**: Food with higher carbohydrates tends to have larger MAE error. **Right**: Performance disparities across different countries in NUTRIBENCH.

hydrate content of single-item meals in NUTRIBENCH. We focus on single-food item meals for this analysis to avoid confounding the results with the effect of increasing food items in the meals. Our analysis reveals a positive correlation: the MAE increases as the carbohydrate content in the meal rises. This suggests that LLM predictions are likely to be more accurate for individuals with a generally low-carbohydrate diet compared to those following a high-carbohydrate diet.

**Performance discrepancy across different countries.** We also measure the performance of LLMs on meal descriptions generated from various countries. Figure 7 (*Right*) displays the MAE for GPT-4o with CoT prompting across different countries in NUTRIBENCH. We observe significant disparities in the performance across these countries, with the lowest error of 2.20 on meals from Nigeria and the highest error of 15.12 on meals from Sri Lanka. Further analysis indicates a correlation between the average carbohydrate content per meal for these countries (blue bars in Figure 7 (*Right*)) and the MAE, with countries with a higher MAE generally tending to contain meals with higher carbohydrates. This pattern indicates that model performance may be influenced by the culinary variety and dietary habits of different cultures, such as the carbohydrate content in meals. Overall, our findings suggest that LLMs perform better for meals associated with certain cultural contexts and cuisines, highlighting the need for more equitable training practices that ensure diverse cultural and dietary representation.

## 5.3 FINE-TUNING WITH FDC DATA

To fine-tune our model, we use the FDC database to convert individual food items into natural language meal descriptions. This process yields a dataset of 600K meal descriptions (details provided

in the Appendix). We then apply the Base prompting method to generate responses for all the meal descriptions, which serves as our training data. We select Gemma2-27B as the base model because it offers similar performance to Llama3.1-405B-FP8 while being significantly smaller in size, making it more suitable for fine-tuning. To fine-tune the model, we utilize qLoRA (Dettmers et al., 2024) with a rank of 8, training for one epoch across 8 A100 GPUs.

Table 3: Fine-tuning the LLM with nutrition data significantly improves MAE (lower is better), accuracy and answer rate (higher is better). Our fine-tuned model is denoted as Gemma2-27B-FT.

| Model
Prompting Method | GPT-4o
CoT | Gemma2-27B
CoT | Gemma2-27B
Base | Gemma2-27B-FT
Base |
|---|---|---|---|---|
| MAE | 8.61 | 13.74 | 13.33 | 9.57 |
| Acc @ 7.5 | 66.82% | 49.35% | 45.61% | 61.71% |
| Answer Rate | 99.16% | 96.79% | 88.65% | 100% |

As shown in Table 3, fine-tuning Gemma2-27B significantly improves performance over pretrained Gemma2-27B by reducing MAE, while also increasing both accuracy and answer rate. However, Gemma2-27B-FT still underperforms GPT-4o with CoT prompting, highlighting GPT-4o's superior capability in carbohydrate estimation. This suggests that fine-tuning models from the GPT family could be a promising direction for future work to achieve an LLM-based nutritionist.

# 6 LLMs ACHIEVE COMPARABLE ACCURACY TO NUTRITIONISTS BUT SIGNIFICANTLY OUTPERFORM IN SPEED

We invited three professional nutritionists in a voluntary study to provide carbohydrate estimates for a subset of meal descriptions from NUTRIBENCH. We sampled 72 meal descriptions, including both natural and metric servings from the U.S. subset, since the nutritionists are based in the United States. For a fair comparison, we explicitly instructed the participants not to search nutrition apps or the web for carbohydrate estimates. We also provided them with three meal descriptions and their carbohydrates, identical to the few-shot examples given in the LLM prompts.

Notably, as shown in Figure 2, we find that several LLMs provide more accurate carbohydrate estimates than the nutritionists for the same 72 queries. Specifically, as detailed in Appendix.L, GPT models excel in estimating carbohydrates for complex, multi-component meals and those with detailed measurements. In contrast, nutritionists perform better when estimating simpler, traditional meals that lack specific brand information. Additionally, the nutritionists report an average of 43 minutes spent estimating responses for all 72 queries. In contrast, LLMs can answer all 72 queries in just a few minutes, with GPT-4o-mini completing them in only 2 minutes.

These findings suggest that LLMs provide accurate carbohydrate estimates that can assist nutritionists in their work, while also significantly reducing the time required to process queries. However, when given access to nutritional lookup resources, nutritionists can achieve results comparable to GPT models, as shown in Appendix.L. This highlights the significant potential of LLMs to enhance the speed and accuracy of nutrition assessments, making them valuable tools for both professionals and non-professionals.

# 7 REAL-WORLD RISK ASSESSMENT

We present a real-world risk assessment demonstrating the impact of nutrition estimation by simulating the effect of meal carbohydrate predictions on the blood glucose levels of individuals with Type 1 diabetes (T1D). To do this, we use the Tidepool Data Science Simulator [3]. The simulator includes a patient metabolism model that emulates how a virtual patient's blood glucose will change over a period of time in response to external events (e.g.- carbohydrate or insulin intake). It also includes a pump simulator running the FDA-cleared Loop algorithm (Tidepool) that delivers insulin to the virtual patient in response to the blood glucose levels and reported carbohydrate intake or additional insulin administration. In real-world scenarios, individuals with Type 1 diabetes either use such pumps or self-administer insulin to manage their blood glucose.

The patient metabolism model is based on three key parameters: the Insulin Sensitivity Factor (ISF), Carb Insulin Ratio (CIR), and the basal rate. We ran simulations for 20 virtual patients using CIR,

---

[3]https://github.com/tidepool-org/data-science-simulator

ISF, and basal rates derived from real, anonymized data from Tidepool. Each simulation modeled a scenario where a virtual T1D patient consumed a meal at the start. The patient's metabolic response was influenced by the actual carbohydrate content of the meal. We then simulated how the patient would manage glucose using estimated carbohydrates provided by nutritionists or LLMs to calculate insulin doses or input into their pump. Two scenarios were explored for each simulation: (1) the patient, not using a pump, calculates their insulin dose manually based on estimated carbs 'x' and their CIR (using the equation x/CIR), and (2) the patient, using a pump, enters estimated carbs into the pump, which adjusts the insulin dose automatically.

We simulated scenarios where patients obtained carbohydrate estimates for meals from nutritionists and GPT-4o. To capture a wide range of real-world conditions, we also varied the patient's blood glucose at meal intake, setting it at 80, 120, 160, and 200 mg/dL. In total, across 20 virtual patients, 4 carbohydrate estimation methods, 4 starting

Table 4: Simulation results: Carbohydrate estimates by GPT-4o with CoT prompting lead to the highest %TIR, lowest %TBR, and lowest BGRI

| Carb Estimator | %TIR | %TBR | %TAR | BGRI |
|---|---|---|---|---|
| Nutritionist 1 | 66.97 | 8.79 | **24.23** | 24.01 |
| Nutritionist 2 | 66.88 | 6.76 | 26.36 | 22.48 |
| Nutritionist 3 | 65.93 | 7.64 | 26.43 | 20.81 |
| GPT-4o (CoT) | **69.88** | **4.18** | 25.94 | **17.83** |

glucose levels, 2 simulation setups, and 70 meals, we conducted 44,800 simulations. Each simulation ran for 6 hours, capturing the following blood glucose metrics:

- **%Time in Range (%TIR):** Percentage of time blood glucose remained within the safe range of 70-180 mg/dL.

- **%Time Below Range (%TBR):** Percentage of time below 70 mg/dL.

- **%Time Above Range (%TAR):** Percentage of time above 180 mg/dL.

- **Blood Glucose Risk Index (BGRI):** A combined measure of high and low blood glucose risk, calculated using the blood glucose risk curve defined by Clarke & Kovatchev (2009).

Table 4 presents the metrics across the different simulation scenarios based on meal carbohydrate predictions from nutritionists and LLMs. We find that carbohydrate estimates made by GPT-4o with CoT prompting result in the highest TIR and the lowest TBR. Although Nutritionist 1's estimates lead to the lowest TAR, low blood glucose poses a greater medical risk than higher levels (Clarke & Kovatchev, 2009). Accordingly, the overall blood glucose risk is significantly lower with GPT-4o's estimates compared to both nutritionists. Figure 8 shows simulated glucose traces for a pump-using patient consuming a meal at t = 0. Predictions by Nutritionists 1-3 and GPT-4o (with CoT prompting) are used in the four scenarios. We observe that while glucose remains within the safe range for

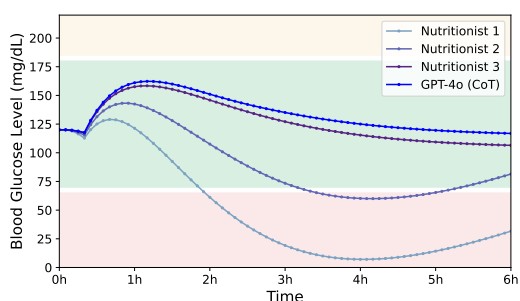

Figure 8: Simulated glucose traces for a virtual patient using a pump with carbohydrate estimates from GPT-4o and nutritionists. The safe glucose range (70-180 mg/dl) is in green, low (<70 mg/dl) in red, and high (>180 mg/dl) in yellow.

simulations with GPT-4o and Nutritionist 3, those involving Nutritionists 1 and 2 drop below range, particularly for Nutritionist 1.

This case study indicates that LLMs can be valuable tools in real-life healthcare settings, helping to prevent risky outcomes and promote better health.

## 8 CONCLUSION

We present NUTRIBENCH, the first publicly available benchmark for evaluating the performance of LLMs in nutrition estimation from natural language meal descriptions. NUTRIBENCH comprises 11,857 human-verified meal descriptions, generated from real-world dietary intake data across 11 countries. We conducted extensive experiments to evaluate twelve state-of-the-art LLMs on carbohydrate estimation, with GPT-4o with CoT prompting achieving the highest accuracy, even surpassing professional nutritionists in both accuracy and speed.

## 9 REPRODUCIBILITY STATEMENT

We describe our data creation process in Section 3, with the serving size conversion explained in Appendix F. The prompts used for meal generation are provided in Appendix B and for LLM evaluation are provided in Appendix C. The details of constructing the RAG database and fine-tuning data are provided in Appendix D and Appendix E, respectively.

## 10 ACKNOWLEDGMENT

We extend our sincere gratitude to Xuan Yang and Yifan Wei for the valuable initial discussions, data survey, and construction efforts for this project. We are also grateful to Xuezhi Wang for the insightful discussion and interpretation of our results. We further thank Naveen Mysore for participating in discussions and developing a demo. Finally, we thank Janice Macleod, Sandra Chmelnik, Susan R. McClendon for providing perspectives and nutrition estimates for the human study as a professional nutritionist. We especially appreciate Susan R. McClendon for providing results for both the look-up and non-look-up conditions and for engaging in discussions. Their collective contributions were essential to the success of our work.

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

## A  MORE EXAMPLES OF GENERATED MEAL DESCRIPTIONS

In this section, we give more examples of meal descriptions in NUTRIBENCH.

### A.1  MEAL DESCRIPTION GENERATED FROM WWEIA

1. For my snack, I poured myself a glass of refreshing white table wine.
   *carbohydrates - 4.68g, energy - 147.6kcal, protein - 0.13g, fat - 0g*

2. For dinner, I feasted on a McDonald's Big Mac, enjoyed a large fast food order of French fries, and sipped on a large drink of caffeine-free fruit-flavored soft drink.
   *carbohydrates - 194.17g, energy - 1394.25kcal, protein - 31.02g, fat - 55.56g*

3. For my evening meal, I savored a fresh regular bagel with a slice of American cheese melted on top.
   *carbohydrates - 56.86g, energy - 341.67kcal, protein - 14.47g, fat - 6.23g*

4. Tonight's dinner consisted of a hearty 230g serving of macaroni noodles in a rich cheese sauce.
   *carbohydrates - 50.46g, energy - 363.40kcal, protein - 11.04g, fat - 12.72g*

5. During my snack time, I enjoyed 150g of ripe raw peach, complemented by 126g of banana and a crunchy 28g serving of baked potato chips.
   *carbohydrates - 63.92g, energy - 306.46kcal, protein - 4.13g, fat - 5.92g*

6. For a satisfying snack, I savored a 152g turnover that was filled with a scrumptious combination of meat and cheese and topped with a tomato-based sauce.
   *carbohydrates - 46.63g, energy - 383.04kcal, protein - 14.17g, fat - 15.55g*

### A.2  MEAL DESCRIPTION GENERATED FROM FAO/WHO GIFT

1. For lunch today, I had a nutritious serving of 0.5 cup of raw lentils and 0.5 cup of crispy fried meat substitute cubes. *(Argentina)*
   *carbohydrates - 105.53g, energy - 599.99kcal, protein - 32.99g, fat - 6.30g*

2. This morning, I enjoyed a raw quince, slicing half of the fruit to savor its crisp texture and aromatic flavor. *(Italy)*
   *carbohydrates - 7.04g, energy - 26.22kcal, protein - 0.18g, fat - 0.05g*

3. For breakfast, I savored a mixture of half a cup of mung beans and a quarter cup of fresh coconut, providing a wholesome and energizing meal. *(Sri Lanka)*
   *carbohydrates - 68.05g, energy - 434.37kcal, protein - 25.40g, fat - 8.31g*

4. At dinner, I savored 100g of brown wheat bread complemented by 350g of rich quince compote. *(Bulgaria)*
   *carbohydrates - 104.40g, energy - 466.00kcal, protein - 8.70g, fat - 1.80g*

5. At lunchtime, I indulged in 466g of leavened bread, made with common millet and similar grains. *(Ethiopia)*
   *carbohydrates - 178.50g, energy - 764.00kcal, protein - 19.10g, fat - 3.30g*

6. For my snack, I had 110g of raw common guavas and 123.75g of fresh mango. *(India)*
   *carbohydrates - 34.29g, energy - 149.05kcal, protein - 3.82g, fat - 1.52g*

## B  DETAILS ON MEAL DESCRIPTION GENERATION

In this section, we provide details about the generation of meal descriptions, including the prompts used for creating these descriptions and human verification examples.

### B.1  PROMPTS FOR MEAL DESCRIPTION GENERATION

We show the prompt used for generating meal descriptions in Figure 9 and the prompt used for combining meal descriptions in fine-tuning section in Figure 10.

**Prompts for GPT-4o-mini to generate meal descriptions, natural serving:**

For the given food item along with their serving size and eating occasion, create a meal description, mimicking natural language. Generate five diverse meal descriptions for each food item. Here are some examples:

Example input- Food list: {{"description": ["smoothie, made with spinach, banana, almond milk, protein powder"], "unit": ["1.0 glass"], "eating_occasion": "Breakfast"}}
Example output- My breakfast usually consists of a glass of smoothie made with spinach, banana, almond milk, and protein powder.

Example input- Food list: {{"description": ["Beverages, NESTEA, tea, black, ready-to-drink, lemon", "Fruit butters, apple"], "unit": ["1.0 serving (8 fl oz)", "1.0 tbsp"], "eating_occasion": "Snack"}}
Example output- For my snack, I enjoyed an 8 fl oz serving of NESTEA lemon black tea along with a tablespoon of apple butter.

Example input- Food list: {{"description": ["Apples, dehydrated (low moisture), sulfured, uncooked", "Alcoholic beverage, beer, regular, BUDWEISER", "Fruit butters, apple"], "unit": ["1.0 cup", "1.0 serving (12 fl oz)", "1.0 tbsp"], "eating_occasion": "Lunch"}}
Example output- For lunch, I had a cup of uncooked dehydrated apples, a tablespoon of apple butter, and a serving of Budweiser beer.

Example input- Food list: {{"description": ["Breakfast tart, lowfat", "Cereal (General Mills Lucky Charms)", "Milk, whole", "Apple juice, 100%"], "unit": ["1 Pop Tart", "1 prepackaged bowl", "1 cup", "1 individual school container"], "eating_occasion": "Breakfast"}}
Example output- For breakfast, I had a low-fat breakfast tart, a prepackaged bowl of General Mills Lucky Charms cereal, a cup of whole milk, and an individual school container of apple juice.

Example input- Food list: {{"description": ["Fruit flavored drink", "Peanut butter sandwich, with regular peanut butter, on white bread", "Corn chips, plain (Fritos)", "Yogurt, low fat milk, flavors other than fruit", "Butterfinger"], "unit": ["1 Little Hug bottle", "1 sandwich", "1 small single serving bag", "1 tube", "1 fun size bar"], "eating_occasion": "Lunch"}}
Example output- For lunch, I enjoyed a Little Hug fruit-flavored drink, a peanut butter sandwich on white bread, a small bag of Fritos, a tube of low-fat yogurt, and a fun-size Butterfinger.

Return only the meal descriptions without additional information. Provide the five descriptions separated by a new line.
Food list: {food_list}

**Prompts for GPT-4o-mini to generate meal descriptions, metric serving:**

For the given food item along with their serving size and eating occasion, create a meal description, mimicking natural language. Generate five diverse meal descriptions for each food item. Here are some examples:

Example input- Food list: {{"description": ["smoothie, made with spinach, banana, almond milk, protein powder"], "unit": [200.0g], "eating_occasion": "Breakfast"}}
Example output- My breakfast usually consists of 200g smoothie made with spinach, banana, almond milk, and protein powder.

Example input- Food list: {{"description": ["Beverages, NESTEA, tea, black, ready-to-drink, lemon", "Fruit butters, apple"], "unit": ["245.0g", "17.0g"], "eating_occasion": "Snack"}}
Example output- My snack today was 245 grams of NESTEA black tea with lemon and a small 17g portion of apple butter.

Example input- Food list: {{"description": ["Apples, dehydrated (low moisture), sulfured, uncooked", "Alcoholic beverage, beer, regular, BUDWEISER", "Fruit butters, apple"], "unit": ["60.0g", "357.0g", "17.0g"], "eating_occasion": "Lunch"}}
Example output- For lunch, I had 60g uncooked dehydrated apples, 17g of apple butter, and 357 of Budweiser beer.

Example input- Food list: {{"description": ["Breakfast tart, lowfat", "Cereal (General Mills Lucky Charms)", "Milk, whole", "Apple juice, 100%"], "unit": ["52.0g", "27.0g", "244.0g", "124.0g"], "eating_occasion": "Breakfast"}}
Example output- For breakfast, I had 52g low-fat breakfast tart, 27g General Mills Lucky Charms cereal, 244g whole milk, and 124g apple juice.

Example input- Food list: {{"description": ["Fruit flavored drink", "Peanut butter sandwich, with regular peanut butter, on white bread", "Corn chips, plain (Fritos)", "Yogurt, low fat milk, flavors other than fruit", "Butterfinger"], "unit": [248.0g", "92.0g", "28.0g", "64.0g", "21.0g"], "eating_occasion": "Lunch"}}
Example output- For lunch, I enjoyed 248g fruit-flavored drink, 92g peanut butter sandwich on white bread, 28g Fritos, 64g low-fat yogurt, and 21g Butterfinger.

Return only the meal descriptions without additional information. Provide the five descriptions separated by a new line.
Food list: {food_list}

Figure 9: Prompts for meal description generation.

## B.2 HUMAN VERIFICATION

After generating five meal descriptions with GPT-4o-mini for each food item, we randomly choose one meal description as the final meal description. However, these descriptions still require refinement before practical application. Here are some examples of the raw generated descriptions and the descriptions after human verification:

**Food descriptions:** *['Boiled eggs, SOURCE-COMMODITIES = Hen eggs', 'Salt, PROCESS = Boiling', 'Vegetable fats and oils, edible, PROCESS = Boiling', 'Strawberries', 'Cow milk, whole', 'White sugar', 'Tortilla'].*
**Serving size:** *['47.8g', '0.2g', '3.9g', '36.2g', '172.6g', '10.1g', '69.2g']*

**Prompts for GPT-4o-mini to combine meal descriptions with natural serving:**

Combine two meal descriptions into one, creating a natural-sounding sentence as if spoken by a person. If two queries describe different meals, merge them into a single meal description.

Example queries: ['For breakfast, I am having a large banana.', 'My lunch consists of a piece of refrigerated whole wheat naan bread.']
Example output: For lunch, I am having a large banana and a piece of refrigerated whole wheat naan bread.

Example queries: ['I had a fresh green salad with a slice of garlic bread on the side.', 'For lunch, I had one serving of grilled chicken with steamed vegetables.', 'I enjoyed a bowl of tomato soup for lunch.']
Example output: For lunch, I enjoyed a fresh green salad with a slice of garlic bread, followed by a hearty bowl of tomato soup and one serving of grilled chicken with steamed vegetables.

Example queries: ['I had a classic burger with cheddar cheese and crispy bacon', 'Enjoyed one serving of seasoned fries', 'For breakfast, I am having a side of coleslaw', 'I am having a cup of chocolate milk for drink.']
Example output: I had a classic burger with cheddar cheese and crispy bacon, served with one serving of seasoned fries, a side of coleslaw and a cup of chocolate milk for breakfast.

Return only the combined meal descriptions, excluding any additional information.

Queries: {query}

**Prompts for GPT-4o-mini to combine meal descriptions with metric serving:**

Combine two meal descriptions into one, creating a natural-sounding sentence as if spoken by a person. If two queries describe different meals, merge them into a single meal description.

Example queries: ['For breakfast, I am having 136g banana.', 'My lunch consists of 106g refrigerated whole wheat naan bread.']
Example output: For lunch, I am having 136g banana and 106g refrigerated whole wheat naan bread.

Example queries: ['I had 43g garlic bread.', 'For lunch, I had 100g grilled chicken with steamed vegetables.', 'I enjoyed 100g tomato soup for lunch.']
Example output: For lunch, I enjoyed 43g garlic bread, followed by 100g tomato soup and 100g grilled chicken with steamed vegetables.

Example queries: ['I had 100g classic burger with cheddar cheese and crispy bacon', 'Enjoyed 85g seasoned fries', 'For breakfast, I am having 220g coleslaw', 'I am having 100g chocolate milk for drink.']
Example output: I had 100g classic burger with cheddar cheese and crispy bacon, served with 85g seasoned fries, 220g coleslaw and 100g chocolate milk for breakfast.

Return only the combined meal descriptions, excluding any additional information.

Queries: {query}

Figure 10: Prompts for meal description combination in fine-tuning.

**Before human verification:** For a refreshing snack, I enjoyed 36.2g of strawberries alongside 47.8g of boiled eggs drizzled with 3.9g of vegetable oil. To complement the eggs, I added a touch of 10.1g of white sugar to sweeten my cup of whole cow milk. Additionally, I savored a tortilla, making for a delightful and satisfying snack experience.
**After human verification:** I enjoyed 47.8g boiled eggs, 3.9g vegetable, and a 69.2g tortilla with 0.2g salt and 10.1g sugar. In addition, I hvae 36.2g strawberries and 172.6g cow milk.
**Changes**: Include missing Foods: salt and cow milk.

**Food descriptions:** *['Pancake syrup', 'Pancakes, with fruit, from frozen'].*
**Serving size:** *['1 tablespoon', '1 pouch']*
**Before human verification:** I kicked off my day with a delightful breakfast of pancakes from frozen, complemented by a tablespoon of rich pancake syrup.
**After human verification:** I kicked off my day with a delightful breakfast of one pouch of pancakes from frozen, complemented by a tablespoon of rich pancake syrup.
**Changes**: Include the missing serving size: one pouch.

**Food descriptions:** *['Chicken fillet biscuit, from fast food', 'Soft drink, cola'].*
**Serving size:** *['1 sandwich, any size', '1 bottle (20 fl oz)']*
**Before human verification:** I savored a chicken fillet biscuit from a fast food joint for breakfast, accompanied by a 20 fl oz bottle of cola.
**After human verification:** I savored a chicken fillet biscuit sandwich from a fast food joint for breakfast, accompanied by a 20 fl oz bottle of cola.
**Changes**: Include the missing serving size: a sandwich.

**Food descriptions:** *['Cream, half and half, flavored', 'Coffee, brewed', 'Banana, raw'].* **Serving size:** *['1 individual container (.5 fl oz)', '1 large', '1 banana']*
**Before human verification:** This morning, my breakfast consisted of a rich brewed coffee, enhanced with a flavored half and half cream, and I couldn't resist having a whole banana on the side.
**After human verification:** This morning, my breakfast consisted of a large brewed coffee, enhanced with an individual container of flavored half and half cream, and I couldn't resist having a whole banana on the side.
**Changes**: Correct the wrong serving size: a large coffee and an individual container.

## C    DETAILS ABOUT EVALUATING LLMS ON CARBOHYDRATE ESTIMATION.

### C.1    HYPERPARAMETERS.

We use $temperature = 0.6$, and $top\_p = 0.9$ for open-sourced models as they are default generating hyperparameters for LLama3.1. For GPT models, we use $top\_p = 0.1$ and $temperature = 0.1$.

### C.2    PROMPTS FOR CARBOHYDRATES ESTIMATION

For Base and CoT methods, we query the LLM model with the meal description and some instructions as shown in Figure 11. For RAG, we will first parse the food description into food components. The parsing prompt is shown in Figure 12. Next, we will retrieve the nutrition information about each food item and finally provide LLMs with the retrieved context as well as the meal description. The RAG prompt for GPT-4o/GPT-4o-mini is shown in Figure 13. Prompts for other models, such as Llama and Gemma, follow their specific templates, though the content remains the same.

## D    CONSTRUCTION OF RAG DATABASE

We constructed RETRI-DBfrom FDC (USDA, 2019) database, using the latest available version at the time of construction (April 2024). The full dataset included 2,046,761 food item entries, of which 489,534 had unique food descriptions. We combined the entries with identical descriptions and filtered those without carbohydrate information. For entries with the same description but varying carbohydrate content, extreme outliers (defined as those with a $z$-score $> 1$) were removed, and the median of the remaining values was taken as the final carbohydrate value for each food item. This process resulted in a final set of 450,182 food items with carbohydrate labels, including 13,048 items with natural serving sizes. Since models process unstructured information more effectively than tabular data (Chen et al., 2019), we applied a rule-based transformation to convert each entry's nutritional context into natural language. Specifically, for an entry containing a *serving amount* and *nutrition amount* for a particular *nutrient*, we convert it to the string "*serving amount* has *nutrition amount nutrient*".

## E    CONSTRUCTION OF FINE-TUNING DATA

We also leverage the FDC data to generate our fine-tuning dataset. We apply a rule-based transformation to convert each food item into a meal description similar to the process in building RETRI-DB. In addition, we sample 98K food items, incorporating both natural and metric servings, and use GPT-4o-mini to generate meal descriptions for these items. To include multiple foods in descriptions, we randomly select 2 to 4 descriptions and use GPT-4o-mini to combine them into a single meal description. The prompt for combining meal descriptions is in Appendix B.1. As a result, we obtain a dataset of 158,246 GPT-generated meal descriptions and 450,182 rule-based meal descriptions.

**Base, GPT-4o-mini/GPT-4o:**
For the given query including a meal description, calculate the amount of carbohydrates in grams. If the serving size of any item in the meal is not specified, assume it is a single standard serving based on common nutritional guidelines (e.g., USDA).
Respond with a dictionary object containing the total carbohydrates in grams as follows:
{{"total_carbohydrates": total grams of carbohydrates for the serving}}
For the total carbohydrates, respond with just the numeric amount of carbohydrates without extra text. If you don't know the answer, respond with:
{{"total_carbohydrates": -1}}.

Query: "This morning, I had a cup of oatmeal with half a sliced banana and a glass of orange juice."
Answer: {{"total_carbohydrates": 66.5}}

Query: "I ate scrambled eggs made with 2 eggs and a toast for breakfast"
Answer: {{"total_carbohydrates": 15}}

Query: "Half a peanut butter and jelly sandwich."
Answer: {{"total_carbohydrates": 25.3}}

Query: {query}
Answer:

---

**CoT, GPT-4o-mini/GPT-4o:**
For the given query including a meal description, think step by step as follows:
1. Parse the meal description into discrete food or beverage items along with their serving size. If the serving size of any item in the meal is not specified, assume it is a single standard serving based on common nutritional guidelines (e.g., USDA). Ignore additional information that doesn't relate to the item name and serving size.
2. For each food or beverage item in the meal, calculate the amount of carbohydrates in grams for the specific serving size.
3. Respond with a dictionary object containing the total carbohydrates in grams as follows:
{{"total_carbohydrates": total grams of carbohydrates for the serving}}
For the total carbohydrates, respond with just the numeric amount of carbohydrates without extra text. If you don't know the answer, set the value of "total_carbohydrates" to -1.

Follow the format of the following examples when answering

Query: "This morning, I had a cup of oatmeal with half a sliced banana and a glass of orange juice."
Answer: Let's think step by step.
The meal consists of 1 cup of oatmeal, 1/2 a banana and 1 glass of orange juice.
1 cup of oatmeal has 27g carbs.
1 banana has 27g carbs so half a banana has (27*(1/2)) = 13.5g carbs.
1 glass of orange juice has 26g carbs.
So the total grams of carbs in the meal = (27 + 13.5 + 26) = 66.5
Output: {{"total_carbohydrates": 66.5}}

Query: "I ate scrambled eggs made with 2 eggs and a toast for breakfast."
Answer: Let's think step by step.
The meal consists of scrambled eggs made with 2 eggs and 1 toast.
Scrambled eggs made with 2 eggs has 2g carbs.
1 toast has 13g carbs.
So the total grams of carbs in the meal = (2 + 13) = 15
Output: {{"total_carbohydrates": 15}}

Query: "Half a peanut butter and jelly sandwich."
Answer: Let's think step by step.
The meal consists of 1/2 a peanut butter and jelly sandwich.
1 peanut butter and jelly sandwich has 50.6g carbs so half a peanut butter and jelly sandwich has (25.3*(1/2)) = 25.3g carbs
So the total grams of carbs in the meal = 25.3
Output: {{"total_carbohydrates": 25.3}}

Query: {query}
Answer: Let's think step by step.

Figure 11: Prompts for carbohydrate estimation using GPT-4o-mini/GPT-4o without RAG.

# F  RULE-BASED ALGORITHM FOR OBTAINING NATURAL SERVING UNITS IN THE FAO/WHO GIFT DATA

For converting the metric servings (grams) for food items in the FAO/WHO GIFT data to natural servings, we first mapped the food items in the FAO/WHO GIFT dataset to those in the FDC dataset by identifying the nearest neighbors in the OpenAI text-embedding-3-large embedding space and choosing a similarity score threshold of 0.5 to obtain valid mappings. A manual inspection of 100 sampled food items showed an accuracy of 85.5% with this threshold. For food items with valid FDC mappings, the serving amount in grams was converted to the nearest natural serving size. This was done by selecting the closest natural serving if it was within 10% of the actual gram amount. If no suitable match was found, the closest larger natural serving was chosen, along with a multiple

> **Parse, GPT-4o-mini:**
> For the given query including a meal description, parse the meal into a list of discrete food or beverage items. Ensure that all the items are strings enclosed in double quotes.
>
> Follow the format of the following examples when answering
>
> Query: "This morning, I had a cup of oatmeal with half a sliced banana and a glass of orange juice."
> Answer: ["oatmeal", "banana", "orange juice"]
>
> Query: "I ate scrambled eggs made with 2 eggs and a toast for breakfast."
> Answer: ["scrambled eggs", "toast"]
>
> Query: "Half a small peanut butter and jelly sandwich."
> Answer: ["peanut butter and jelly sandwich"]
>
> Query: "A large cup of cappuccino made with whole milk"
> Answer: ["cappuccino made with whole milk"]
>
> Query: {query}
> Answer:

Figure 12: Prompts for parsing meal description into food items using GPT-4o-mini/GPT-4o.

Table 5: Mean Absolute Error (MAE) and Acc@7.5 of models fine-tuned using Llama and Gemma, comparing performance across different parameter sizes.

|                   | Mean Absolute Error | Acc@7.5 |
|-------------------|---------------------|---------|
| Llama3.1-8B-FT    | 13.79               | 46.84   |
| Llama3.1-70B-FT   | 12.60               | 49.61   |
| Gemma2-27B-FT     | 10.49               | 56.71   |
| Llama3.1-8B-Base  | 19.97               | 36.20   |
| Llama3.1-70B-Base | 14.73               | 42.05   |
| Gemma2-27B-Base   | 13.32               | 45.61   |

to reflect the correct quantity. To avoid unrealistic servings (e.g., 10 tbsp of rice), if no multiple smaller than 3 was found, we selected the closest smaller serving and expressed it using fractions (e.g., 'quarter tablespoon', 'half a cup'). For even smaller amounts, we applied volume conversions, such as 1/8 of a cup being equivalent to 2 tablespoons or 1/16 of a cup to 1 teaspoon.

## G ADDITIONAL RESULTS ON FINE-TUNING WITH FDC DATA.

To explore whether a larger model or a different model will yield similar performance improvements with fine-tuning, we also conduct ablation studies by fine-tuning LLaMA 3.1 models with 8B and 70B parameters. Due to computational constraints, fine-tuning is limited to GPT-generated data, comprising 39,745 metric samples and 19,745 natural samples. The results, presented in Table 5, indicate that larger model sizes generally lead to better performance. Furthermore, Gemma2 consistently outperforms LLaMA 3.1, aligning with the trends observed in the un-fine-tuned models.

## H EXAMPLES FROM SOURCE DATABASES TO NUTRIBENCH

### H.1 EXAMPLE 1

The raw WWEIA and FNDDS data with selected rows and columns are in Table 6 and 7. The generated meal description is: "At dinner, I treated myself to a delicious double cheeseburger from McDonald's paired with a delightful soft serve vanilla ice cream cone in a waffle cone." The associated nutritional values are: carbohydrates - 112.96g, energy - 957.45kcal, protein - 39.45g, and fat - 38.86g. All nutritional data are sourced from WWEIA (refer to Table 6).

### H.2 EXAMPLE 2

Table 8 presents an example of a meal found in the FAO/WHO Gift raw data for India. The final query generated from this log was 'For my snack, I had 30g of food industry prepared biscuits,

**RAG, GPT-4o-mini:**
Context: {context}
For the given query including a meal description, calculate the amount of carbohydrates in grams. If the serving size of any item in the meal is not specified, assume it is a single standard serving based on common nutritional guidelines (e.g., USDA).
Respond with a dictionary object containing the total carbohydrates in grams as follows:
{{"total_carbohydrates": total grams of carbohydrates for the serving}}
For the total carbohydrates, respond with just the numeric amount of carbohydrates without extra text. If you don't know the answer, respond with:
{{"total_carbohydrates": -1}}.

Query: "This morning, I had a cup of oatmeal with half a sliced banana and a glass of orange juice."
Answer: {{"total_carbohydrates": 66.5}}

Query: "I ate scrambled eggs made with 2 eggs and a toast for breakfast"
Answer: {{"total_carbohydrates": 15}}

Query: "Half a peanut butter and jelly sandwich."
Answer: {{"total_carbohydrates": 25.3}}

Query: {query}
Answer:

---

**RAG+CoT, GPT-4o-mini:**
Context: {context}
For the given query including a meal description, think step by step as follows:
1. Parse the meal description into discrete food or beverage items along with their serving size. If the serving size of any item in the meal is not specified, assume it is a single standard serving based on common nutritional guidelines (e.g., USDA). Ignore additional information that doesn't relate to the item name and serving size.
2. For each food or beverage item in the meal, calculate the amount of carbohydrates in grams for the specific serving size.
3. Respond with a dictionary object containing the total carbohydrates in grams as follows:
{{"total_carbohydrates": total grams of carbohydrates for the serving}}
For the total carbohydrates, respond with just the numeric amount of carbohydrates without extra text. If you don't know the answer, set the value of "total_carbohydrates" to -1.

Follow the format of the following examples when answering

Query: "This morning, I had a cup of oatmeal with half a sliced banana and a glass of orange juice."
Answer: Let's think step by step.
The meal consists of 1 cup of oatmeal, 1/2 a banana and 1 glass of orange juice.
1 cup of oatmeal has 27g carbs.
1 banana has 27g carbs so half a banana has (27*(1/2)) = 13.5g carbs.
1 glass of orange juice has 26g carbs.
So the total grams of carbs in the meal = (27 + 13.5 + 26) = 66.5
Output: {{"total_carbohydrates": 66.5}}

Query: "I ate scrambled eggs made with 2 eggs and a toast for breakfast."
Answer: Let's think step by step.
The meal consists of scrambled eggs made with 2 eggs and 1 toast.
Scrambled eggs made with 2 eggs has 2g carbs.
1 toast has 13g carbs.
So the total grams of carbs in the meal = (2 + 13) = 15
Output: {{"total_carbohydrates": 15}}

Query: "Half a peanut butter and jelly sandwich."
Answer: Let's think step by step.
The meal consists of 1/2 a peanut butter and jelly sandwich.
1 peanut butter and jelly sandwich has 50.6g carbs so half a peanut butter and jelly sandwich has (25.3*(1/2)) = 25.3g carbs
So the total grams of carbs in the meal = 25.3
Output: {{"total_carbohydrates": 25.3}}

Query: {query}
Answer: Let's think step by step.

Figure 13: Prompts for carbohydrate estimation using GPT-4o-mini/GPT-4o with RAG.

| SEQN (Interview ID) | DR1_030Z (Meal Occasion) | DR1IFDCD (Food Code) | DR1IGRMS (Food Weight) | DR1ICARB (Carbohydrate) |
|---|---|---|---|---|
| 100705 | 3.0 (dinner) | 27510387 | 165 | 29.65 |
| 100705 | 3.0 (dinner) | 13120786 | 255 | 83.31 |

Table 6: Selected raw data from WWEIA. This table focuses solely on carbohydrates, while other nutritional information, though omitted here, is available in the database.

paired with 26g of sweet bars made from palm trunk sap and sugar cane molasses, along with 23g of raw peanuts.', with labels 373kcal for energy, 7.5g for protein, 48.9g for carbohydrates, and 17.6g for fat.

| Food Code | Main food description | Portion weight (g) | Portion description |
|---|---|---|---|
| 27510387 | Double cheeseburger (McDonalds) | 165.0 | 1 double cheeseburger |
| 13120786 | Ice cream cone, soft serve, vanilla, waffle cone | 255.0 | 1 cone |

Table 7: Selected raw data from FNDDS

| MEAL_NAME | FOODEX2_INGR_DESCR | FOOD_AMOUNT_CONS | CARBOH_g |
|---|---|---|---|
| 8 | Biscuits, PREPARATION-PRODUCTION-PLACE = Food industry prepared | 30 | 20.4 |
| 8 | Sweet bars and other formed sweet masses, INGREDIENT = Palms (trunk sap), INGREDIENT = Sugar cane molasses, PHYSICAL-STATE = Solid (soft or hard) | 26 | 24.8 |
| 8 | Peanuts, PROCESS = Raw, no heat treatment | 23 | 3.7 |

Table 8: Example of a meal in the raw data of FAO/WHO Gift from India. Several columns are omitted for space considerations. The column MEAL_NAME notes the eating occasion (breakfast, lunch, dinner etc.), FOODEX2_INGR_DESCR refers to the food items consumed in the meal, FOOD_AMOUNT_CONSUMED provides the portion size in grams of the consumed food item, and CARBOH_g indicates the corresponding carbohydrates.

## I  ANALYSIS OF HIGH CARB VS. LOW CARB MEALS

We analyze the properties of single-food item high vs low-carb meals further to investigate the higher error rates in the single-food high-carb meals observed in Section 5.2. We defined high-carb meals as single food item meals with carbohydrate values above the third quartile (33.93g, N=869). Similarly, low-carb meals were the single food item meals with carbohydrate values below the first quartile (6.47g, N=871).

We found that high-carb meals exhibited significantly greater variability in carbohydrate content compared to low-carb meals ($P < 0.05$, Levene's test). Specifically, the standard deviation for high-carb meals was 26.88g, whereas for low-carb meals, it was only 2.01g. This increased variance in high-carb meals suggests that the model may face greater difficulty when predicting carbohydrate content for foods with a broader range of values, which likely contributes to the observed increase in error rates.

Additionally, we observed significant differences in the portion weights of low-carb and high-carb meals. High-carbohydrate meals had larger average portion sizes (95.27 ± 139.89 g) compared to low-carbohydrate meals (226.80 ± 161.73 g) ($P < 0.05$, Mann-Whitney U Test). The larger portion sizes of high-carb meals could also increase the complexity of accurate carbohydrate estimation, as larger meals may involve more varied ingredient types or require more precise portioning, contributing to the higher error rates.

## J  ERROR ANALYSIS

We conducted a detailed analysis of the queries in NutriBench leading to high errors with the best-performing model, GPT-4o with CoT prompting. First, we examined 100 queries sampled from NutriBench where the model exhibited a high absolute error ($>50$g). We found that in all cases analyzed, the model successfully parsed meal components and serving sizes, accurately formulated equations, and correctly performed mathematical operations to calculate the total carbohydrate content. However, it produced inaccurate carbohydrate estimates for one or more items within the meals, leading to overall high errors.

To further identify patterns in the errors, we compared high-error meals (absolute error >third quartile, N=2,940) with low-error meals (absolute error <third quartile, N=2,939) using the Mann-Whitney U Test. The analysis revealed that high-error meals contained more food items on average ($2.13 \pm 0.93$) compared to low-error meals ($1.81 \pm 0.83$, $P < 0.05$). High-error meals also had significantly higher carbohydrate content ($64.58 \pm 38.69$g vs. $24.13 \pm 25.27$g, $P < 0.05$) as well as larger portion weights ($268.37 \pm 250.08$g) compared to low-error meals ($437.28 \pm 301.25$g, $P < 0.05$).

These findings highlight that meal complexity, greater carbohydrate content, and larger portion sizes are key challenges for LLMs in carbohydrate estimation that should be focused on in future work to improve model performance.

## K    EVALUATING WITH PARAPHRASED PROMPTS

To evaluate the robustness of NutriBench to variations in prompt phrasing, we paraphrased the instructions while retaining specific output formats and demonstrations. This approach ensures consistency in expected outputs while testing the model's flexibility in understanding rephrased instructions.

The three paraphrased instructions were as follows:

1. Given a query describing a meal, estimate the total carbohydrates in grams. If the serving size of any food item is not mentioned, assume a standard single serving based on common nutritional guidelines (e.g., USDA). Provide the result as a dictionary in the following format: {{"total_carbohydrates": total grams of carbohydrates}} Include only the numeric carbohydrate value without additional text. If the information needed to calculate the value is missing or uncertain, respond with: {{"total_carbohydrates": -1}}

2. Analyze the provided meal description and calculate the total carbohydrate content in grams. If the portion size of any ingredient is not specified, assume a default single serving based on standard nutritional guidelines (e.g., USDA). Return the result as a dictionary in the format: {{"total_carbohydrates": total grams of carbohydrates}} Ensure that only the numeric value is included for the total carbohydrates. If the necessary information is unavailable or unclear, reply with: {{"total_carbohydrates": -1}}

3. Estimate the total carbohydrates in grams for a given meal description. If the portion size for any food item is not specified, assume a standard single serving size based on common nutritional guidelines (e.g., USDA). Provide the result as a dictionary in this format: {{"total_carbohydrates": total grams of carbohydrates}}

   Use only the numeric carbohydrate value in your response without any additional text. If necessary information to compute the value is missing or ambiguous, respond with: {{"total_carbohydrates": -1}}

| Experiment | Acc@7.5 | Mean Absolute Error |
| --- | --- | --- |
| Instruction 1 (original) | 51.43 | 11.46 |
| Instruction 2 | 51.29 | 11.54 |
| Instruction 3 | 52.70 | 11.78 |
| Instruction 4 | 50.75 | 11.56 |
| **Mean** | **51.54** | **11.59** |
| **Standard Deviation** | **0.71** | **0.11** |
| **p-value** | **0.80** | **0.16** |

Table 9: Performance metrics for different instructions.

The performance of GPT-4o-mini with Base prompt is summarized in Table 9. While the prompt does not need to adhere rigidly to a specific formulation to produce results, it must maintain a defined structure, including clear output formats and demonstrations, to guide the model effectively. Paraphrasing the prompt has minimal impact on the results.

## L  MORE ANALYSIS ABOUT GPT VS. NUTRITIONISTS

Among 72 meal descriptions, we identify 20 queries where GPT outperforms all nutritionists, and 8 meal descriptions where all nutritionists outperform GPT. Our analysis reveals intriguing patterns:

- **GPT excels in complex, multi-component meals and those with detailed measurements**. For instance, in the description "For breakfast, I had a Burger King sandwich featuring egg, cheese, and sausage on a biscuit, paired with a can of cola," GPT achieves a Mean Absolute Error (MAE) of 6.09, compared to the lowest MAE of 10.09 among nutritionists.
- **Nutritionists perform better with simpler, traditional meals lacking specific brand information.** For example, in the description "Tonight's dinner consisted of a hearty 230g serving of macaroni noodles in a rich cheese sauce,". GPT has an MAE of 20.84, while the highest MAE among nutritionists is 10.46.

Since we only have carbohydrate estimations from nutritionists, it is difficult to directly evaluate their knowledge of specific meal descriptions. However, by analyzing the variance in their estimations, we uncover interesting patterns:

- For meal descriptions with the highest variance among nutritionist estimations, GPT achieves a substantially lower Mean Absolute Error (MAE) of 18.9, compared to the lowest MAE of 34.4 among nutritionists (averaged over the top 10 high-variance descriptions).
- For meal descriptions with the lowest variance, the MAEs are much closer: GPT achieves an MAE of 4.6, while nutritionists' MAEs range from 3.4 to 4.5.

These findings suggest that GPT performs better on meal descriptions where nutritionists show greater disagreement, possibly due to gaps in their knowledge or unfamiliarity with the meals.

The primary reason we did not initially allow nutritionists to search online was to prevent them from accessing our source database and obtaining the ground truth values directly. However, after discussions with nutritionists, we learned that they may rely on tools to look up information as part of their usual workflow. To ensure a fair comparison, we conduct an additional human study where nutritionists are allowed to look up food items and use their standard methods to estimate carbohydrates.

In this study, we received only one result from a nutritionist, which achieved comparable performance to the best AI model, CoT GPT-4o. Compared to the scenario where nutritionists are not allowed to look up information, we observe two significant improvements: Meal descriptions with metric servings show better performance, as demonstrated in Table 10. Meal descriptions where nutritionists previously disagreed improved significantly, with the MAE of the top 10 high-variance descriptions dropping from 36.7 to 21.4, although GPT still performed better overall.

| Method | Acc@7.5, all | Acc@7.5, metric | Acc@7.5, natural |
|---|---|---|---|
| Nutritionist | 42.45 | 39.47 | 45.45 |
| Nutritionist, look up | 59.72 | 73.68 | 44.12 |
| GPT-4o, CoT | 60.56 | 63.16 | 57.58 |

Table 10: Performance metrics allowing nutritionists to look up.

