# OpenReview forum: "NutriBench: A Dataset for Evaluating Large Language Models in Nutrition Estimation from Meal Descriptions"
_ICLR.cc/2025/Conference — ICLR 2025 Poster_

### Official Review · Reviewer_bDdB · 2024-10-31

**Soundness:** 3
**Presentation:** 3
**Contribution:** 3
**Rating:** 6
**Confidence:** 5

**Summary:**

This paper describes a new dataset, NutriBench, that contains over 10,000 synthetically generated natural language meal descriptions and corresponding macronutrients from matching foods in the Food Data Central database. They run experiments with 12 open and proprietary large language models, including varients of GPT, Qwen, and Llama on the task of carbohydrate prediction. They found that GPT-4o with Chain-of-Thought prompting resulted in the highest accuracy and response rate.

**Strengths:**

The greatest strength is the public release of the NutriBench dataset. Particularly impressive is that the dataset is based on real meals that people across 11 different countries eat. Another strength is the extensive experiments that were run, testing 12 LLMs, using three prompting strategies, comparing to nutritionists, and simulating the effect of carbohydrate predictions on blood glucose levels of individuals with diabetes. The takeaways for which LLMs perform best under which conditions (i.e., prompt, natural vs metric serving size, and diet/country) are important for researchers in this area. The real-world risk assessment with the Tidepool simulation of blood sugar levels is interesting, as are the LLM prompts in the appendix. Finally, this is an important problem with broad societal impacts—diet tracking is challenging, and finding ways to reduce user burden is critical for continued usage, which will help with obesity, diabetes, heart disease, and many other health conditions.

**Weaknesses:**

The biggest strength is the database, but the data that will be released and was included in the supplementary material contains only the natural language meal descriptions, no Food Data Central entities or their nutrition information. In addition, the only task was carbohydrate prediction, and the authors mention that the dataset contains only macronutrient information, but an LLM nutritionist should be able to predict protein and fat as well, and micronutrients as very important too. 11,857 meal descriptions is a small number, but is reasonable given they are human-verified. Another major weakness is the study in section 6 comparing LLMs to human nutritionists. It is understandable that human experts are slower, but instructing nutritionists not to search the web for carbohydrate estimates and then claiming that LLMs are more accurate seems quite problematic and results in a very misleading conclusion. Nutritionists do their job by using tools and resources. You can’t get an accurate measure of their ability to estimate carbohydrates if you take away the tools they use in the real world. The only actual takeaway here is that LLMs are faster, which is obvious. I love this work and really wanted to accept this paper, but can’t justify it due to these weaknesses.

**Questions:**

•	Note that Answer Rate and RETRI-DB are used before they are defined—consider moving up the definitions to the first place these terms are used.
•	Consider explaining why you used generative AI to write the natural language meal descriptions instead of humans.
•	Were all the foods eaten by people in 11 countries in the USDA’s Food Data Central? I’m surprised you had full coverage of globally eaten meals just from the US food database.
•	Please fix the caption for Figures 4 and 5.
•	In section 5.2, you comment that LLM predictions are more accurate for individuals on a low-carb diet. However, aren’t people on a high-carb diet more likely to get diabetes?
•	In section 5.3, why did you use the Base prompting method instead of CoT to generate responses when you said yourself that CoT is more accurate?
•	In real life, when someone has diabetes, how do they dose their insulin typically? I’m guessing they’re not calculating their carbs every time they eat.
•	In Appendix B.2, a couple of the human-verified responses actually seemed worse to me, with misspellings and unnecessary food words added (“sandwich” to “biscuit”).
•	Why is your FDC database only 450K foods? Doesn’t it contain over 1.8M foods?

---

> ### Author Response · Authors · 2024-11-22
>
> Thank you for recognizing our paper's strengths and providing constructive feedback. We truly appreciate your enthusiasm for our work. We have worked to address the noted weaknesses and incorporate the suggestions to improve our paper further.
>
> > The biggest strength is the database, but the data that will be released and was included in the supplementary material contains only the natural language meal descriptions, no Food Data Central entities or their nutrition information.
>
> Thank you for raising this point. **We have updated the supplementary material to include macronutrient and calorie labels for each meal description.** We will also update the submission with a public link to the dataset after the review process. Additionally, we have added examples in Appendix H showing entries from the raw databases with the final meal descriptions and nutrition labels in NutriBench.
>
> > In addition, the only task was carbohydrate prediction, and the authors mention that the dataset contains only macronutrient information, but an LLM nutritionist should be able to predict protein and fat as well, and micronutrients as very important too.
>
> While we perform a detailed study of carbohydrate estimation with LLMs using the NutriBench dataset, we will publicly release the dataset with macronutrient labels (proteins, carbohydrates, and fats), as well as calorie information to support further research in this area. We hope that NutriBench can serve as a benchmark to inspire future studies on a variety of applications in this area, including meal and health planning, virtual nutritionists, dietary recommendations, and calorie tracking for conditions such as obesity and heart disease, where accurate nutrition estimation is a critical component.
> We currently do not include micronutrient information in the dataset due to its limited availability in the source datasets (WWEIA and FAO/WHO GIFT), leaving this as an opportunity for future work.
>
> > It is understandable that human experts are slower, but instructing nutritionists not to search the web for carbohydrate estimates and then claiming that LLMs are more accurate seems quite problematic and results in a very misleading conclusion.
>
> Thank you for your question. The primary reason we initially instructed nutritionists not to search online was to prevent them from directly accessing our source database and obtaining ground truth values. Additionally, we wanted to evaluate human performance by professionals possessing domain expertise and knowledge about food and nutrition.
> However, after discussions with nutritionists, we recognized that using tools to look up information is part of their usual workflow. To ensure a fair comparison, we conducted an additional human study where nutritionists were allowed to look up food items and use their standard methods to estimate carbohydrates.
>
> Due to the short time frame, we were able to obtain estimates for this follow-up study from one nutritionist, who showed comparable accuracy to the best model, GPT-4 with CoT prompting, as shown in the table below. Metric/Natural refers to meal descriptions with specified portion sizes, such as '100g' for metric or '1 cup' for natural measurements.
>
> | Model                 | Acc@7.5, all | Acc@7.5, metric | Acc@7.5, natural |
> |-----------------------|--------------|------------------|------------------|
> | Nutritionist, no look up          | 42.45\%        | 39.47\%           | 45.45           |
> | Nutritionist, allow look up | 59.72\%        | 73.68\%           | 44.12\%           |
> | GPT-4o, CoT          | 60.56\%        | 63.16\%           | 57.58\%           |
>
> We have also included an analysis in the Appendix L comparing scenarios where GPT outperforms nutritionists and vice versa. However, we note that while look-up tools improve human accuracy, they significantly increase the time taken to answer queries(1 hour 33 minutes allowing look-up vs 41 minutes wihout look-up).
>
> Overall, our findings show that LLMs can provide nutritional estimates with accuracy comparable to nutritionists using external lookup tools, but in a fraction of the time. This highlights their potential for integration into standard workflows to enable fast and precise nutritional assessments, effectively supporting nutritionists in their daily tasks. We have updated the claim in Section 6 accordingly and will include the finalized details of the follow-up study.

---

> > ### Author Response · Authors · 2024-11-22
> >
> > > Note that Answer Rate and RETRI-DB are used before they are defined—consider moving up the definitions to the first place these terms are used.
> >
> > Thank you for noting this- we have revised the paper to ensure that the terms are only used after they are defined.
> >
> > > Consider explaining why you used generative AI to write the natural language meal descriptions instead of humans.
> >
> > Thank you for your suggestion. We chose GPT-4o-mini to generate the meal descriptions due to the size of the dataset, especially as we continue to scale and expand the benchmark. We have added this explanation in Section 3.3.
> >
> > > Were all the foods eaten by people in 11 countries in the USDA’s Food Data Central? I’m surprised you had full coverage of globally eaten meals just from the US food database.
> >
> > The meal descriptions for countries outside the United States were **sourced from the FAO/WHO GIFT database**. As this dataset provides food measurements in grams, we converted portions to natural serving sizes by mapping food items to the USDA’s FDC database. We also included 261 meals from the FAO/WHO GIFT dataset where at least one food item could not be mapped to the FDC database with a high similarity score, to minimize potential biases and ensure comprehensive representation.
> >
> > We also conducted a manual inspection of the FDC database and found coverage of a diverse range of global food items. Examples include Ajweh (a traditional Arabic food), Caribbean-style plantain veggie burger patties, Kerala mixture (an Indian snack), Surasang Octopus Dumplings (a South Korean branded food), and Moo Shu pork (a Chinese dish).
> >
> > > Please fix the caption for Figures 4 and 5.
> >
> > We have revised the captions for Figures 4 and 5 to provide more descriptive explanations of the plots.
> >
> > > In section 5.2, you comment that LLM predictions are more accurate for individuals on a low-carb diet. However, aren’t people on a high-carb diet more likely to get diabetes?
> >
> > Thank you for your question. We consulted one of our collaborators, an Assistant Professor of Medicine (Endocrinology), and an expert in the field of diabetes. According to their expertise, individuals who benefit most from such carb-counting tools are typically those with type 1 diabetes, which is an autoimmune disorder unrelated to carbohydrate intake. For individuals with type 2 diabetes, the risk is more strongly associated with excess calorie consumption (in any form), along with factors such as obesity and genetics.
> >
> > > In section 5.3, why did you use the Base prompting method instead of CoT to generate responses when you said yourself that CoT is more accurate?
> >
> > Thank you for your question. Our initial fine-tuning experiments used the base prompting method as a way to provide the model with nutritional information and to teach it to process natural language meal queries. While this approach significantly improved the model's performance compared to the non-finetuned version, we observed that when fine-tuned using training data formatted with base prompts, **the model lost its ability to effectively apply CoT reasoning**. As a result, outputs generated with the CoT prompt became nearly identical to those generated with the base prompt.
> >
> > To investigate the impact of including CoT reasoning in the training data itself, we generated synthetic reasoning traces using a rule-based algorithm for parsing meal descriptions, estimating the carbohydrates for each food item in the meal, and computing the total carbohydrates as the overall estimate. We applied this approach to a subset of the training data. The table below compares the performance of Gemma2-27B fine-tuned with this CoT data versus the same subset of data but in the baseline prompt format. **We find that the model fine-tuned with CoT prompt performs well when tested with the CoT prompt.** While these findings are promising, improving fine-tuning methods to fully leverage CoT reasoning remains an area for future work.
> >
> > | Model   | Fine-Tuning Data | Prompting Method | Acc@7.5 |
> > |--------|---------------------------|-----------------------|--------------------------|
> > | Gemma2-27B | None             | Baseline             | 45.61\%                   |
> > | Gemma2-27B | None             | CoT             | 49.35\%                   |
> > | Gemma2-27B | Baseline             | Baseline             | 56.71\%                   |
> > | Gemma2-27B | CoT             | CoT             | 63.07\%                   |

---

> > > ### Author Response · Authors · 2024-11-22
> > >
> > > > In real life, when someone has diabetes, how do they dose their insulin typically? I’m guessing they’re not calculating their carbs every time they eat.
> > >
> > > Thank you for your question. According to our clinician collaborator, carb counting is necessary for optimal glycemic control and is required for insulin delivery systems to perform effectively. This also ensures consistency, allowing healthcare providers to make accurate adjustments to treatment plans. Currently, even the most advanced commercial insulin delivery systems rely on carb counting.
> > > Existing diabetes literature also emphasizes that carb counting is essential for adjusting the prandial insulin bolus to match the carb content of each meal. This is because carbohydrates are the primary macronutrient influencing postprandial glucose levels [^1].
> > > Additionally, one of the authors has been a diabetes patient for over 10 years and personally follows the practice of carb counting for every meal.
> > >
> > > [^1]: [Amorim, D.; Miranda, F.; Santos, A.; Graça, L.; Rodrigues, J.; Rocha, M.; Pereira, M.A.; Sousa, C.; Felgueiras, P.; Abreu, C. Assessing Carbohydrate Counting Accuracy: Current Limitations and Future Directions. Nutrients 2024, 16, 2183. https://doi.org/10.3390/nu16142183](https://www.mdpi.com/2072-6643/16/14/2183)
> > >
> > > > In Appendix B.2, a couple of the human-verified responses actually seemed worse to me, with misspellings and unnecessary food words added (“sandwich” to “biscuit”).
> > >
> > > Thank you for your feedback. We verified that the misspelling was introduced only in the paper while transcribing the example from the data and does not exist in the actual dataset. This has been corrected in the manuscript. Regarding the second example, the original food entry used to generate the meal description was “Chicken fillet biscuit, from fast food.” and the portion size was “1 sandwich, any size”. To preserve the recorded food item and the portion size, we corrected it to "a chicken fillet biscuit sandwich". We have also updated the examples provided in Appendix B with explanations of the changes made and the original data entries from which the meal descriptions were generated for clarity.
> > >
> > > > Why is your FDC database only 450K foods? Doesn’t it contain over 1.8M foods?
> > >
> > > Yes, while the FDC database contains a total of 2,046,761 entries, only 489,534 of these food descriptions are unique. Many entries with the same food name have either duplicate or varying nutritional content. For our purposes, we combined entries sharing the same descriptions and filtered those without carbohydrate information. Further, we removed extreme outlier carbohydrate values (z-score > 1) among entries with the same name and calculated the median of the remaining entries as the final carbohydrate content. This process resulted in a final dataset of 450,182 unique food items with carbohydrate data, which was then used to construct the RAG database. We have updated the section discussing the RAG database construction (Appendix D) to include more details of the data processing steps and raw data counts.
> > >
> > >
> > >
> > > We hope our responses and clarifications have addressed your concerns. If you feel the issues have been resolved, we would greatly appreciate it if you could consider updating your score. Thank you for your valuable time and thoughtful feedback.

---

> > > ### Comment · Reviewer_bDdB · 2024-11-22
> > >
> > > Thank you for your responses. I'm curious when you say you "also included 261 meals from the FAO/WHO GIFT dataset where at least one food item could not be mapped to the FDC database with a high similarity score," if it could not be mapped to the FDC database, where did you get the ground truth nutrition facts?

---

> > > > ### Author Response · Authors · 2024-11-22
> > > >
> > > > Thank you for raising the score based on our responses and the follow-up question! We would like to clarify that the ground truth nutritional information for the meals in the FAO/WHO GIFT dataset was included in the dataset itself. **The FDC database was only used to facilitate serving size conversions from grams to natural units**.An example of a query generated from the FAO/WHO GIFT dataset is provided in Appendix H.2. We would be happy to provide any additional clarifications or details you might need!

---

> > ### Comment · Reviewer_bDdB · 2024-11-22
> >
> > This is excellent, thank you for making all those changes, updating the dataset to include nutrient information, and running more experiments, including a study with one nutritionist who was allowed to look up nutrition facts. In your dataset, I would recommend including the identified food database entries and the nutrition facts for each so it is clear how you went from the full natural language meal description to the final nutrition facts. I agree, for the camera-ready paper, you will need to run a proper study with more than 1 nutritionist.

---

### Official Review · Reviewer_TX8E · 2024-11-02

**Soundness:** 3
**Presentation:** 3
**Contribution:** 3
**Rating:** 6
**Confidence:** 3

**Summary:**

The paper introduces a benchmark called NutriBench, which includes 11,857 meal descriptions derived from real-world global dietary intake data. It employs Chain of Thought (CoT) and Retrieval-Augmented Generation (RAG) techniques to tackle the carbohydrate estimation task and evaluates twelve large language models (LLMs), such as GPT-4o and Llama 3.1. Ultimately, a real-world carbohydrate prediction task is conducted. The results demonstrate a significant improvement in both the accuracy and speed of carbohydrate estimation compared to traditional nutritionists.

**Strengths:**

1. Data Collection: This study gathers 11,857 meal descriptions from various countries. This approach enhances text flexibility compared to traditional tabular data and addresses the challenges of acquiring image data.
2. Model Evaluation: The research evaluates the carbohydrate prediction capabilities of different large language models (LLMs). The results show that these models outperform nutritionists in both prediction speed and accuracy.
3. Social Influence: The paper conducts a real-world risk assessment by simulating how carbohydrate predictions affect blood glucose levels in individuals with diabetes. This demonstrates the potential positive impact of LLMs on public health.

**Weaknesses:**

1. Limited downstream tasks: the paper only focuses on the carbohydrate prediction task. More downstream tasks can be performed, e.g., protein prediction, etc.
2. Over-reliance on description quality: the prediction performance generated by LLMs relies heavily on the quality (comprehensiveness, accuracy, etc) of description data.

**Questions:**

More downstream tasks and comprehensive analyses on nutrition can be performed to generate more informative insights, compared to just providing carbohydrate predictions.

---

> ### Author Response · Authors · 2024-11-22
>
> Thank you for acknowledging our strengths and providing your valuable feedback. We hope our response adequately addresses your concerns.
>
> > Limited downstream tasks: the paper only focuses on the carbohydrate prediction task. More downstream tasks can be performed, e.g., protein prediction, etc. More downstream tasks and comprehensive analyses on nutrition can be performed to generate more informative insights, compared to just providing carbohydrate predictions.
>
> Thank you for your question. We focus on carbohydrate estimation as one of the tasks made possible to evaluate with NutriBench, which is critical for diabetes management as carb counting plays an essential role in adjusting prandial insulin doses and maintaining safe blood glucose levels [^1]. Beyond carbohydrate estimation, this benchmark can facilitate future research on applications like meal and health planning, virtual nutritionists, dietary recommendations, and calorie tracking for conditions such as obesity and heart disease, where accurate nutrition estimation is a critical component. While these tasks are beyond the scope of this paper, we have made the dataset publicly available to encourage further exploration and development in these directions.
>
> [^1]: [Amorim, D.; Miranda, F.; Santos, A.; Graça, L.; Rodrigues, J.; Rocha, M.; Pereira, M.A.; Sousa, C.; Felgueiras, P.; Abreu, C. Assessing Carbohydrate Counting Accuracy: Current Limitations and Future Directions. Nutrients 2024, 16, 2183. https://doi.org/10.3390/nu16142183](https://www.mdpi.com/2072-6643/16/14/2183)
>
>
>
> > Over-reliance on description quality: the prediction performance generated by LLMs relies heavily on the quality (comprehensiveness, accuracy, etc) of description data.
>
> We conducted human verification for all the meal descriptions in the benchmark, particularly focusing on correcting inaccuracies related to food items and servings in the meals. We relied on FAO/WHO, WWEIA, and FNDDS datasets as the ground truth sources and made sure that all the information needed to make accurate nutrition estimates (e.g. inclusion of ingredients/food components, serving sizes, etc.) was included in the final query in the human verification step. Further details and examples of the human verification process are provided in Section 3.3 and Appendix B.2.

---

> > ### Comment · Reviewer_TX8E · 2024-11-23
> >
> > Thank you for the clarifications, your replies make sense to me.

---

### Official Review · Reviewer_rcnA · 2024-11-04

**Soundness:** 3
**Presentation:** 3
**Contribution:** 3
**Rating:** 6
**Confidence:** 3

**Summary:**

This study presented NutriBench, a natural language meal description dataset labeled with macronutrient and calorie estimates. It evaluated 12 LLMs covering both open and closed source models with different prompting strategies on carbohydrate estimation. It also conducted a study with 3 nutritionists to obtain carbohydrate estimates on a sample of meal descriptions. Through a real-world risk assessment by simulating the effect of carbohydrate estimates on the blood glucose levels of Type 1 diabetes patiens, it found that LLM can help Type 1 diabetes patients maintain their blood glucose within safe limits.

**Strengths:**

(1) This study worked on an important issue which may influence diabetes management.

(2) Most of the writings are clear.

(3) In addition to a benchmark, this study also provided a preliminary experiments comparing LLMs with human experts. It also showed how it might be applied to real-world management via a case study.

**Weaknesses:**

(1) The study might be much improved if it could provide more insights into the success and failure cases of the LLMs on the carbohydrate estimation task. When LLMs made wrong estimations, what were those mistakes? Was it because LLMs did not have the information, or it was more like a math operation issue. To me, it is unclear what capabilities of LLMs did this study intend to find out. If it is just for evaluating LLMs for carbohydrate estimation, the research contribution might be limited. It would be better if the authors could further explain how the findings of this study may inspire future works. Why is it different from other LLM benchmark papers?

(2) The study should disclose hyperpoparameter settings in detail.

**Questions:**

Questions:

(1) NutriBench consists of 11,858 meal descriptions. Was the human verification applied to all of them?

(2) It appears that human verification might induce subjective biases. Take the pizza case as an example, why should we update "a tasty medium crust pepperoni pizza" to "a piece of tasty medium crust pepperoni pizza"?

(3) Is the benchmark robust and reproducible? Even though the temperature can be set 0 to obtain more deterministic responses, it is uncertain whether the responses will be reproducible. It would be better to include additional experiments for this concern.

(4) Does the prompt have to be strictly formulated to produce the results? Would paraphrasing the prompt significantly change the results?

(5) Would there be any biases when the data generator and carbohydrate estimator are identical? In this case, both are GPT-4o-mini.

(6) What is the x-axis in Figure 6?

(7) In the simulation experiment, it seems that the performance of different nutritionist varies significantly. Why is that?

(8) According to Figure 2, the human nutritionist performed worse than a lot of LLMs. When human made mistakes, what were those mistakes? Did LLMs perform better just because they held more knowledge? If each food is associated with a pre-defined value, why can't nutritionists looked those up if they were not sure of the knowledge?

(9) The authors discussed that previous benchmarks mainly focused on using images. I wonder whether it is feasible to caption the images from these datasets and expand the existing one. Additionally, why not caption the images instead of constructing a benchmark from the scratch?

Minor:

(1) It would be better to use verb tense consistently.

---

> ### Author Response · Authors · 2024-11-22
>
> Dear Reviewer rcnA,
> Thank you for your throughtful comments and valuable feedback. We hope our response can address your concerns.
>
> > The study might be much improved if it could provide more insights into the success and failure cases of the LLMs on the carbohydrate estimation task. When LLMs made wrong estimations, what were those mistakes? Was it because LLMs did not have the information, or it was more like a math operation issue.
>
> Thank you for your insightful question. To investigate, we manually analyzed 100 queries from NutriBench where the best-performing model, GPT-4o with chain-of-thought prompting, had a high absolute error (>50g). In all cases, the model successfully parsed meal components and serving sizes, correctly formulated equations, and performed mathematical operations to determine the total carbohydrate content. However, **it made inaccurate carbohydrate estimates** for one or more items within the meals. We further compared high-error meals (absolute error > third quartile, N=2,940) with low-error meals (absolute error < third quartile, N=2,939) using the Mann-Whitney U Test. We found that **high-error meals were more complex,** with a higher average number of food items (2.13 ± 0.93 vs. 1.81 ± 0.83, P < 0.05), higher carbohydrate content (64.58 ± 38.69g vs. 24.13 ± 25.27g, P < 0.05), and larger portion weights (437.28 ± 301.25g vs. 268.37 ± 250.08g, P < 0.05). These results offer insights into the types of meals that are challenging for LLMs and should be an area of focus in future work. We have updated Appendix J in the paper with this analysis.
>
> > To me, it is unclear what capabilities of LLMs did this study intend to find out. If it is just for evaluating LLMs for carbohydrate estimation, the research contribution might be limited. It would be better if the authors could further explain how the findings of this study may inspire future works. Why is it different from other LLM benchmark papers?
>
> Thank you for your question. This study introduces **the first and only benchmark with natural language meal descriptions annotated with macronutrients (proteins, carbohydrates, and fats) and calories**. This benchmark is designed to facilitate research in the broad area of nutrition with LLMs and supports a variety of downstream applications.
>
> One key focus of this paper is the task of carbohydrate estimation, which is critical for diabetes management as carb counting plays an essential role in adjusting prandial insulin doses and maintaining safe blood glucose levels [^1]. By evaluating LLMs on this task, our research highlights their current capabilities, such as providing carbohydrate estimates that are more accurate or comparable to those of seasoned nutritionists. **This demonstrates their potential as valuable tools for both patients and professional nutritionists.** At the same time, we identify areas for improvement, such as addressing discrepancies in performance across cultural diets and food items from different countries.
>
> Beyond carbohydrate estimation, this benchmark can facilitate future research on applications like meal and health planning, virtual nutritionists, dietary recommendations, and calorie tracking for conditions such as obesity and heart disease. While these tasks are beyond the scope of this paper, we have made the dataset publicly available to encourage further exploration and development in these directions.
>
> [^1]: [Amorim, D.; Miranda, F.; Santos, A.; Graça, L.; Rodrigues, J.; Rocha, M.; Pereira, M.A.; Sousa, C.; Felgueiras, P.; Abreu, C. Assessing Carbohydrate Counting Accuracy: Current Limitations and Future Directions. Nutrients 2024, 16, 2183. https://doi.org/10.3390/nu16142183](https://www.mdpi.com/2072-6643/16/14/2183)

---

> > ### Author Response · Authors · 2024-11-22
> >
> > > The study should disclose hyperparameter settings in detail. Is the benchmark robust and reproducible? Even though the temperature can be set 0 to obtain more deterministic responses, it is uncertain whether the responses will be reproducible. It would be better to include additional experiments for this concern.
> >
> > For the open-sourced models, we use a temperature of 0.6 and top_p of 0.9, which are the default generating hyperparameters for LLama3.1. For GPT models, we set temperature and top_p to 0.1 to ensure more deterministic outputs. The hyperparameter details have been added to Appendix C.1.
> >
> > To address concerns about reproducibility, we ran additional experiments with the “Base, Llama-3.1-8B” model, evaluating the benchmark three times. The results, including those presented in the paper, **demonstrate minimal variance** (standard deviation of absolute error: 0.08) and confirm that the observed differences across runs are **not statistically significant** (p-values computed using the original value for both Mean Absolute Error and Acc@7.5 are greater than 0.05). These findings are summarized in the table below, highlighting the robustness and reproducibility of our benchmark.
> >
> > | Experiment                 | Mean Absolute Error | Acc@7.5 |
> > |---------------------|---------------------|---------|
> > | Run 1 (original)    | 19.97              | 36.20   |
> > | Run 2               | 20.00              | 35.69   |
> > | Run 3               | 19.81              | 36.15   |
> > | Run 4               | 19.93              | 35.44   |
> > | **Mean**            | **19.93**          | **35.87** |
> > | **Standard Deviation** | **0.08**          | **0.36** |
> > | **p-value** | **0.38**          | **0.17** |
> >
> > > NutriBench consists of 11,858 meal descriptions. Was the human verification applied to all of them?
> >
> > Yes, we verify all the queries. We rely on WHO, WWEIA, and FNDDS as  the ground truth and make sure all information needed to accurately answer the question (e.g. inclusion of ingredients/food components, serving sizes, etc.) is included in the query in the human verification step, as discussed in Section 3.3. We also used a rule-based method (exact match search of meal components and portion size) to find incorrect descriptions that may have been missed and corrected them manually. In total, 440/11,858 meal descriptions were manually modified across the entire NutriBench dataset.
> >
> > >  It appears that human verification might induce subjective biases. Take the pizza case as an example, why should we update "a tasty medium crust pepperoni pizza" to "a piece of tasty medium crust pepperoni pizza"?
> >
> > For this example, the original food items in the meal are: *['Orange juice, 100%, canned, bottled or in a carton', 'Pizza, with pepperoni, from school lunch, medium crust']*. And the original serving sizes correponding to each item are: *['1 individual school carton', '1 piece, NFS']*.
> > Before human verification, the query generated by GPT-4o-mini was *"a tasty medium crust pepperoni pizza"*, implying consumption of an entire pizza. **However, this interpretation is inconsistent with the carbohydrate label, which is based on a single piece of pizza as specified in the original food units.** To align the query with the correct unit and carbohydrate value, we manually revised it to *"a piece of tasty medium crust pepperoni pizza."* This ensures the query reflects the intended portion size and accurately matches the nutritional information.
> >
> > > Does the prompt have to be strictly formulated to produce the results? Would paraphrasing the prompt significantly change the results?
> >
> > The prompt **does not have to be strictly formulated** to produce the results, but it must follow a certain structure, including specific output formats and demonstrations, to guide the model effectively. **Simply paraphrasing the prompt does not significantly change the results**, as long as the structure and key elements of the instructions remain intact. We conducted an experiment using three additional paraphrased instructions on GPT-4o-mini and the result is in Table. The paraphrased instructions can be found in the Appendix K.
> >
> > | **Experiment**       | **Acc@7.5** | **Mean Absolute Error** |
> > |-----------------------|-------------|--------------------------|
> > | Instruction 1 (original)      | 51.43       | 11.46                   |
> > | Instruction 2                 | 51.29       | 11.54                   |
> > | Instruction 3                 | 52.70       | 11.78                   |
> > | Instruction 4                 | 50.75       | 11.56                   |
> > | **Mean**              | **51.54**       | **11.59**                   |
> > | **Standard Deviation**| **0.71**        | **0.11**                    |
> > | **p-value**           | **0.80**       | **0.16**                   |

---

> > > ### Author Response · Authors · 2024-11-22
> > >
> > > > Would there be any biases when the data generator and carbohydrate estimator are identical? In this case, both are GPT-4o-mini.
> > >
> > >
> > > Thank you for your question. **We do not anticipate biases to arise from using identical models (GPT-4o-mini) for both the data generator and the carbohydrate estimator.** The data generation process only serves to add natural language context to the meals, while the underlying food items and their nutritional content are derived directly from the WWEIA and WHO/FAO datasets. Further, the human verification step ensures that the generated meal descriptions accurately reflect the items in the original data information.
> > >
> > > > What is the x-axis in Figure 6?
> > >
> > > Thank you for raising this point. The x-axis in Figure 6 represents the answer rate. We have updated the figure in the revised version of the paper.
> > >
> > > > In the simulation experiment, it seems that the performance of different nutritionist varies significantly. Why is that?
> > >
> > > Thank you for raising this question. In Figure 8, we present one simulation for comparing blood glucose traces based on meal carbohydrate counts provided by GPT-4o and three nutritionists. In this specific example, the meal contained 15.88g of carbohydrates. However, the estimates varied significantly: Nutritionists 1 and 2 predicted 48g and 30g, respectively, leading to overestimated insulin doses and subsequent drops in blood glucose. Conversely, Nutritionist 3 and GPT-4o estimated 17g and 14.3g, respectively, resulting in blood glucose remaining within safe limits throughout the simulation.
> > > These variations can also be partly attributed to the patient in this simulation having a relatively high insulin sensitivity factor (134 mg/dL per unit), making their blood glucose highly responsive to small differences in insulin dosing. This amplifies the impact of carbohydrate estimation errors on glucose dynamics.
> > > While this example highlights variability, the overall trends shown in Table 4 indicate that, across all scenarios and patients, the performance of the nutritionists was generally consistent, with no significant variation observed at the aggregate level.
> > >
> > > > According to Figure 2, the human nutritionist performed worse than a lot of LLMs. When human made mistakes, what were those mistakes?
> > >
> > >
> > > Among 72 meal descriptions, we identify 20 queries where GPT outperforms all nutritionists, and 8 meal descriptions where all nutritionists outperform GPT. Our analysis reveals intriguing patterns:
> > >
> > > * **GPT excels in complex, multi-component meals and those with detailed measurements.** For instance, in the description "For breakfast, I had a Burger King sandwich featuring egg, cheese, and sausage on a biscuit, paired with a can of cola," GPT achieves a Mean Absolute Error (MAE) of 6.09, compared to the lowest MAE of 10.09 among nutritionists.
> > > * **Nutritionists perform better with simpler, traditional meals lacking specific brand information.** For example, in the description "Tonight's dinner consisted of a hearty 230g serving of macaroni noodles in a rich cheese sauce," GPT has an MAE of 20.84, while the highest MAE among nutritionists is 10.46.
> > >
> > > > Did LLMs perform better just because they held more knowledge?
> > >
> > > Since we only have carbohydrate estimations from nutritionists, it is difficult to directly evaluate their knowledge of specific meal descriptions. However, by analyzing the variance in their estimations, we uncover interesting patterns:
> > >
> > > * **For meal descriptions with the highest variance among nutritionist estimations,** GPT achieves a substantially lower Mean Absolute Error (MAE) of 18.9, compared to the lowest MAE of 34.4 among nutritionists (averaged over the top 10 high-variance descriptions).
> > > * **For meal descriptions with the lowest variance**, the MAEs are much closer: GPT achieves an MAE of 4.6, while nutritionists' MAEs range from 3.4 to 4.5.
> > >
> > > These findings suggest that GPT performs better on meal descriptions where nutritionists show greater disagreement, possibly due to gaps in their knowledge or unfamiliarity with the meals.

---

> > > > ### Author Response · Authors · 2024-11-22
> > > >
> > > > > If each food is associated with a pre-defined value, why can't nutritionists looked those up if they were not sure of the knowledge?
> > > >
> > > > Thank you for your question. The primary reason we instructed nutritionists not to search online was to prevent them from directly accessing our source database and obtaining the ground truth values.  Additionally, we wanted to evaluate human performance by professionals who possess domain expertise and knowledge about food and nutrition. However, after further discussion with nutritionists, we learned that they may rely on tools to look up information as part of their usual workflow. To ensure a fair comparison, we are conducting an additional human study where nutritionists are allowed to look up food items and use their standard methods to estimate carbohydrates.
> > > >
> > > > Due to the short time frame, we were able to obtain estimates for this study from one nutritionist, who achieved comparable accuracy as the best model, GPT-4o with CoT prompting, as shown in the table below.
> > > >
> > > >
> > > > | Model                 | Acc@7.5, all | Acc@7.5, metric | Acc@7.5, natural |
> > > > |-----------------------|--------------|------------------|------------------|
> > > > | Nutritionist, no look up          | 42.45\%        | 39.47\%           | 45.45           |
> > > > | Nutritionist, allow look up | 59.72\%        | 73.68\%           | 44.12\%           |
> > > > | GPT-4o, CoT          | 60.56\%        | 63.16\%           | 57.58\%           |
> > > >
> > > > Comparing this scenario with the one where nutritionists were not allowed to look up information, we observe two significant improvements:
> > > >
> > > > * **Nutritionist accuracy improves on meal descriptions with metric servings** when allowed to look up external sources.
> > > > * **Meal descriptions where nutritionists previously disagreed improve significantly**, with the MAE of the top 10 high-variance descriptions dropping from 36.7 to 21.4, although GPT still perform better overall.
> > > >
> > > > > The authors discussed that previous benchmarks mainly focused on using images. I wonder whether it is feasible to caption the images from these datasets and expand the existing one. Additionally, why not caption the images instead of constructing a benchmark from the scratch?
> > > >
> > > > Thank you for the question. Captioning images to expand benchmarks is feasible but comes with significant challenges. First, automated captioning models often lack domain-specific precision, leading to errors such as misidentifying foods or overlooking important details. This issue is exacerbated when dealing with foods from different countries, as the models may lack the necessary cultural or regional knowledge. Second, captioning accuracy is affected by variations in angles, lighting, and context, which can result in missed or mislabeled food items, particularly in mixed dishes or when items are partially obscured.
> > > >
> > > > > It would be better to use verb tense consistently.
> > > >
> > > > Thank you for the suggestion. We are working on ensuring consistent verb tense in the next version.
> > > >
> > > > We hope that our responses and clarifications have addressed your concerns. If you feel our explanations have resolved the issues, we kindly encourage you to consider updating your score accordingly. Thank you once again for your valuable time and thoughtful feedback.

---

> ### Author Response · Authors · 2024-11-25
>
> Dear Reviewer rcnA, thank you again for dedicating your time and effort to review our paper. With just under two days remaining, we would love to know if we have adequately addressed your concerns and whether this has influenced your score. We have put a lot of effort into updating our work and would value your feedback. We are also happy to provide any additional clarifications or details you might need!

---

> > ### Comment · Reviewer_rcnA · 2024-11-25
> >
> > I have carefully read the thorough responses and additional experiments. They have addressed my concerns. Therefore, I raised the contribution score to 3 and the overall score to 6.
> >
> > I find the new experimental results very interesting and would like to thank the authors for performing the experiments. I hope they can be incorporated into the future version of the manuscript.
> >
> > Can the authors discuss the future directions of the development of LLMs in this context? How can we further improve the models on nutrition estimation?

---

> > > ### Author Response · Authors · 2024-11-25
> > >
> > > Thank you for your thoughtful feedback and for raising the score. We truly appreciate your suggestions and are glad the additional experiments addressed your concerns; we are working on incorporating them into our manuscript.
> > > Regarding future directions, we have identified that the main limitation of LLMs in nutrition estimation lies in their nutritional knowledge base. To address this, we plan to integrate reliable external knowledge bases and enhance our fine-tuning and RAG approaches. Additionally, we plan to explore modeling nutrition databases as knowledge graphs, which could enable more structured and interpretable reasoning. Another future direction includes incorporating decision-making approaches such as AI agents to ultimately develop a virtual nutritionist capable of providing personalized and accurate nutritional guidance. We believe these efforts will significantly advance the practical applications of LLMs in this field.

---

> > > > ### Comment · Reviewer_rcnA · 2024-11-25
> > > >
> > > > Thank you for the summary. I am curious about the knowledge integration part. Is the knowledge required for nutrition estimation changing constantly and rapidly? What would be a good strategy to construct the dataset, and design the training mechanism if the knowledge changes quickly anyway? Similarly, is there a feasible and efficient way to update the proposed benchmark?
> > > >
> > > > The question regarding knowledge integration may be beyond the scope of the current study. I just wonder if there is any good idea. The rebuttal has already addressed my concerns.

---

> > > > > ### Author Response · Authors · 2024-11-25
> > > > >
> > > > > Thank you for the interesting follow-up question! From our experience with the FDC database which is updated bi-annually, nutrition facts for existing items tend to remain stable, while there may be additions of new items. Therefore the training and knowledge database can be updated with the new knowledge at a similar frequency. Finally, regarding NutriBench, we will update it whenever the source database is updated.

---

### Official Review · Reviewer_JjPd · 2024-11-04

**Soundness:** 3
**Presentation:** 3
**Contribution:** 3
**Rating:** 6
**Confidence:** 4

**Summary:**

This paper presents NUTRIBENCH, a new benchmark dataset designed to evaluate LLMs on nutrition estimation from natural language meal descriptions. The dataset includes 11,857 meal descriptions annotated with macronutrient labels (carbohydrates, proteins, fats, and calories), generated from global dietary intake data. The authors assess twelve LLMs.

**Strengths:**

- Nutribench is the first of its kind for the task of macronutrient estimation from the textual description of meals.
- The paper benchmarks multiple LLMs and also explores/compares multiple prompting strategies.
- The authors conduct an evaluation involving nutritionists as well.
- The authors do a comprehensive analysis of performance across different dimensions.

**Weaknesses:**

- The authors have relied on GPT4o to generate meal descriptions. There is a concern with this. Were all of the outputs by GPT4o verified by humans?
- The fine-tuning experiment with Gemma2-27B shows improvements but does not address whether larger or different models would perform similarly with fine-tuning.
- The choice of specific prompting methods (e.g., why Chain-of-Thought outperforms other methods) should be discussed well.
- Although the paper mentions about the performance disparities across cultural diets, it does not explicitly address if these are due to model biases or data representation gaps or any unique characteristics of the specific foods.

**Questions:**

- What measures are taken to verify the nutritional information accuracy of each meal description, especially considering human verification only involves a single author?
- Could you provide additional details on how you mapped food quantities to everyday serving sizes? For instance, how does the rule-based algorithm handle ambiguous measurements?
- How do you explain the higher error rates in carbohydrate estimation for high-carbohydrate foods? Does the model struggle with certain food types or serving sizes?
- What might be the underlying causes for cultural discrepancies in model performance, particularly for countries like Sri Lanka with higher MAE?

---

> ### Author Response · Authors · 2024-11-22
>
> Dear Reviewer JjPd,
> Thank you for your thoughtful review and valuable feedback. We hope our response can address your concerns.
>
> > What measures are taken to verify the nutritional information accuracy of each meal description, especially considering human verification only involves a single author? Were all of the outputs by GPT4o verified by humans?
>
> The nutrition labels for the meals were directly sourced from the FAO/WHO, WWEIA, and FNDDS datasets. We verified all the meal descriptions generated by GPT-4o-mini to ensure that all the information needed to accurately answer the question (e.g. inclusion of ingredients/food components, serving sizes, etc.) was included in the query in the human verification step, discussed in Section 3.3. We also used a rule-based method (exact match search of meal components and portion size) to find incorrect descriptions that may have been missed and corrected them manually. In total, 440/11,858 meal descriptions were manually modified across the entire NutriBench dataset. Two common mistakes made by GPT-4o-mini were missing food names and missing food servings. As an example of the verification process, the following tables display a meal from the raw WWEIA and FNDDS datasets with selected columns:
>
> | SEQN (Interview ID)   | DR1_030Z (Meal Occasion)| DR1IFDCD (Food Code) | DR1IGRMS (Food Weight)| DR1ICARB (Carbohydrate) |
> |--------|---------------------------|-----------------------|-------------------------|--------------------------|
> | 100705 | 3.0 (dinner)             | 27510387             | 165.0                  | 29.65                   |
> | 100705 | 3.0 (dinner)             | 13120786             | 255.0                  | 83.31                   |
>
> | Food Code | Main Food Description                     | Portion Weight (g) | Portion Description       |
> |-----------|-------------------------------------------|---------------------|---------------------------|
> | 27510387  | Double cheeseburger (McDonalds)          | 165.0              | 1 double cheeseburger     |
> | 13120786  | Ice cream cone, soft serve, vanilla, waffle cone | 255.0              | 1 cone                    |
>
> The generated meal description based on this data is:
>
> &nbsp;&nbsp;&nbsp;&nbsp;&nbsp;&nbsp;&nbsp;&nbsp;&nbsp;&nbsp;"At dinner, I treated myself to a delicious double cheeseburger from McDonald's paired with a delightful soft serve vanilla ice cream cone in a waffle cone."
>
> Both food names ("double cheeseburger" and "soft serve vanilla ice cream") and portion descriptions ("a double cheeseburger," "a cone") are included in this meal description, which meets the criteria for inclusion in NutriBench. We also provide examples of corrections made in the human verification process in Section 3.3 and Appendix B.2.
>
> > The fine-tuning experiment with Gemma2-27B shows improvements but does not address whether larger or different models would perform similarly with fine-tuning.
>
> Thank you for raising this question. To investigate, we additionally fine-tuned LLaMA 3.1 8B and 70B models. Due to computational constraints, we limited fine-tuning to only the GPT-generated data without rule-based meal descriptions described in the Appendix E, specifically 39,745 metric and 19,745 natural samples. The results are shown in Table. **Comparing Llama3.1-8B-FT and Llama3.1-70B-FT, we observe that the larger model leads to improved results. Additionally, Gemma2 outperforms LLaMA 3.1, which is consistent with the results observed in the non-fine-tuned models.** We update this analysis in the Appendix G.
>
> | Model                | Mean Absolute Error | Acc@7.5 |
> |----------------------|---------------------|---------|
> | Llama3.1-8B-FT      | 13.79              | 46.84   |
> | Llama3.1-70B-FT     | 12.60              | 49.61   |
> | Gemma2-27B-FT       | 10.49              | 56.71   |
> | Llama3.1-8B-Base    | 19.97              | 36.20   |
> | Llama3.1-70B-Base   | 14.73              | 42.05   |
> | Gemma2-27B-Base     | 13.32              | 45.61   |
>
> > The choice of specific prompting methods (e.g., why Chain-of-Thought outperforms other methods) should be discussed well.
>
> Thank you for your suggestion. We provide a detailed analysis of the benefits of Chain-of-Thought (CoT) prompting in Section 5.1. Our findings indicate that **CoT particularly reduces errors over baseline prompting for complex queries involving multiple food items.** CoT's step-by-step reasoning helps the model identify meal components and calculate carbohydrate estimates more accurately, a task it struggles with otherwise. We further analyze the performance of RAG in this section, highlighting its dependence on both the model backbone and the data.

---

> > ### Author Response · Authors · 2024-11-22
> >
> > > Could you provide additional details on how you mapped food quantities to everyday serving sizes? For instance, how does the rule-based algorithm handle ambiguous measurements?
> >
> > Thank you for your question. The food records in the FAO/WHO Gift dataset provide serving amounts in grams. To convert these to natural serving sizes, we mapped each food item to the FDC database, which provides conversions between metric and natural serving sizes (e.g., 1 slice of white bread = 8g). To determine the closest natural serving size, we compared the gram weight of each food item to the weight of the natural serving FDC conversions. If the weight was within a 10% threshold of a natural serving size, we used the closest match. Otherwise, we adjusted by selecting the next largest or smallest serving size, using fractions or multiples as needed (e.g., "half a cup" or "2 slices"). Further details of this rule-based algorithm are provided in Appendix F.
> >
> > > How do you explain the higher error rates in carbohydrate estimation for high-carbohydrate foods? Does the model struggle with certain food types or serving sizes?
> >
> > Thank you for the insightful question. We further analyzed properties of single-item low-carb meals (with carbohydrate values below the first quartile, N=871) and high-carb meals (with carbohydrate values above the third quartile, N=869) to complement the analysis of Section 5.2. **We found that high-carb meals exhibit a significantly greater variability**, with a standard deviation of 26.88g compared to just 2.01g for low-carb meals (P < 0.05, Levene's test). This higher variability likely contributes to the observed increase in error rates for high-carbohydrate foods, as the model faces greater challenges when predicting across a broader range of values.
> >
> > Additionally, we observed that **high-carbohydrate meals tend to have significantly larger portion weights** (95.27 ± 139.89 g) compared to low-carbohydrate meals (226.80 ± 161.73g), as determined by the Mann-Whitney U Test (P < 0.05). This suggests that larger portion sizes may also increase the complexity of accurate estimation, further explaining the model's higher error rates for high-carbohydrate foods. We have added this analysis in Appendix I.
> >
> > > Although the paper mentions about the performance disparities across cultural diets, it does not explicitly address if these are due to model biases or data representation gaps or any unique characteristics of the specific foods. What might be the underlying causes for cultural discrepancies in model performance, particularly for countries like Sri Lanka with higher MAE?
> >
> > Thank you for the insightful comments. Several factors could contribute to the performance disparity across cultural diets. However, without detailed knowledge of the training processes for current LLMs, pinpointing the exact sources of this bias is challenging. **One reason we highlighted in our paper is the variation in carbohydrate content across meals in different countries.** In Section 5.2, Figure 7 shows that countries with higher prediction errors often have meals with higher carbohydrate content, which increases prediction error.
> >
> > We also found that **meal portion weights were significantly higher** for meals from Sri Lanka (237.98 ± 169.19g) compared to those from Nigeria (50.43 ± 60.94g), which had the lowest error rate. This difference in portion sizes could be another contributing factor.
> >
> >
> > We hope that our responses and additional clarifications have addressed your concerns. If you find that our explanations satisfactorily resolve the issues you raised, we kindly request you to consider revising your score accordingly. Thank you again for your valuable time and constructive feedback.

---

> ### Author Response · Authors · 2024-11-25
>
> Dear Reviewer JjPd, thank you again for dedicating your time and effort to review our paper. With just under two days remaining, we would love to know if we have adequately addressed your concerns and whether this has influenced your score. We have put a lot of effort into updating our work and would value your feedback. We are also happy to provide any additional clarifications or details you might need!

---

> > ### Comment · Reviewer_JjPd · 2024-11-25
> >
> > Thank you for your response. I’ve revised my scores in light to your responses.

---

> > > ### Author Response · Authors · 2024-11-25
> > >
> > > Thank you for all the comments and raising the score!

---

### Meta-Review · Area_Chair_16MF · 2024-12-17

**Metareview:**

This paper introduces NutriBench, a benchmark dataset with 11,857 natural language meal descriptions annotated with macronutrient data to evaluate LLMs for nutrition estimation. The authors conduct thorough experiments on 12 LLMs, explore prompting strategies, compare predictions with human nutritionists, and perform a real-world risk assessment for Type 1 diabetes management. Strengths include the release of the NutriBench dataset, which is a valuable contribution to nutrition research, the breadth of experiments, and the practical insights on model performance across diverse diets. The paper also addresses a critical problem with societal impact, such as improving diet tracking for health conditions like diabetes and obesity. While the study focuses only on carbohydrate prediction and restricts nutritionist tools in comparison, these limitations do not diminish its overall value. Some methodological concerns regarding comparisons to human nutritionists (e.g., restricting their use of external tools) should be clarified in future work. The NutriBench dataset and findings provide an important resource for future research and applications, justifying acceptance.

**Additional Comments On Reviewer Discussion:**

There are two major discussion points during the rebuttal: one is multiple reviewers asking for additional experiments and information on the experimental designs, as well as results analyses. Authors have done a great job addressing these issues. The other one is the limitation of the variety of downstream tasks. Authors explained why this is the case and the potential of generalizing the results to other applications.

---

### Decision · Program_Chairs · 2025-01-22

Accept (Poster)